# FCoReBench: Can Large Language Models Solve Challenging First-Order Combinatorial Reasoning Problems?

## Abstract

Can the large language models (LLMs) solve challenging first-order combinatorial reasoning problems such as graph coloring, knapsack, and cryptarithmetic? By first-order, we mean these problems can be instantiated with potentially an infinite number of problem instances of varying sizes. They are also challenging being NP-hard and requiring several reasoning steps to reach a solution. While existing work has focused on coming up with datasets with hard benchmarks, there is limited work which exploits the first-order nature of the problem structure. To address this challenge, we present FCoReBench, a dataset of 40 such challenging problems, along with scripts to generate problem instances of varying sizes and automatically verify and generate their solutions. We first observe that LLMs, even when aided by symbolic solvers, perform rather poorly on our dataset, being unable to leverage the underlying structure of these problems. We specifically observe a drop in performance with increasing problem size. In response, we propose a new approach, SymPro-LM, which combines LLMs with both symbolic solvers and program interpreters, along with feedback from a few solved examples, to achieve huge performance gains. Our proposed approach is robust to changes in the problem size, and has the unique characteristic of not requiring any LLM call during inference time, unlike earlier approaches. As an additional experiment, we also demonstrate SymPro-LM's effectiveness on other logical reasoning benchmarks.

## 1 Introduction

Recent works have shown that large language models (LLMs) can reason like humans (Wei et al., 2022a), and solve diverse natural language reasoning tasks, without the need for any fine-tuning (Wei et al., 2022c; Zhou et al., 2023; Zheng et al., 2023). We note that, while impressive, these tasks are simple reasoning problems, generally requiring only a handful of reasoning steps to reach a solution.

We are motivated by the goal of assessing the reasoning limits of modern-day LLMs. In this paper, we study computationally intensive, first-order combinatorial problems posed in natural language. These problems (e.g., sudoku, knapsack, graph coloring, cryptarithmetic) have long served as important testbeds to assess the intelligence of AI systems (Russell and Norvig, 2010), and strong traditional AI methods have been developed for them. Can LLMs solve these directly? If not, can they solve these with the help of symbolic AI systems like SMT solvers? To answer these questions, we release a dataset named FCoReBench, consisting of 40 such problems (see Figure 1).

We refer to such problems as *fcore* (first-order combinatorial reasoning) problems. *Fcore* problems can be instantiated with any number of instances of varying sizes, e.g., 9×9 and 16×16 sudoku. Most of the problems in FCoReBench are NP-hard and solving them will require extensive planning and search over a large number of combinations. We provide scripts to generate instances for each problem and verify/generate their solutions. Across all problems we generate 1354 test instances of varying sizes for evaluation and also provide 596 smaller sized solved instances as a training set. We present a detailed comparison with existing benchmarks in the related work (Section 2).

Not surprisingly, our initial experiments reveal that even the largest LLMs can only solve less than a third of these instances. We then turn to recent approaches that augment LLMs with tools for better reasoning. Program-aided Language models (PAL) (Gao et al., 2023) use LLMs to generate programs,

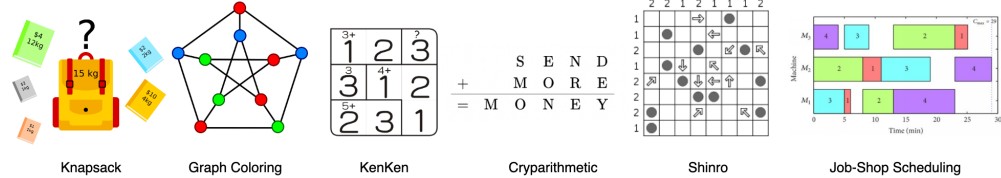

Figure 1: Illustrative examples of problems in FCoReBench (represented as images for illustration).

offloading execution to a program interpreter. Logic-LM (Pan et al., 2023) and SAT-LM (Ye et al., 2023) use LLMs to convert questions to symbolic representations, and external symbolic solvers perform the actual reasoning. Our experiments show that, by themselves, their performances are not that strong on FCoReBench. At the same time, both these methods demonstrate complementary strengths – PAL can handle first-order structures well, whereas Logic-LM is better at complex reasoning. In response, we propose a new approach named SymPro-LM, which combines the powers of *both* PAL and symbolic solvers with LLMs to effectively solve *fcore* problems. In particular, the LLM generates an instance-agnostic program for an *fcore* problem that converts any problem instance to a symbolic representation. This program passes this representation to a symbolic solver, which returns a solution back to the program. The program then converts the symbolic solution to the desired output representation, as per the natural language instruction. Interestingly, in contrast to LLMs with symbolic solvers, once this program is generated, inference on new *fcore* instances (of any size) can be done *without* any LLM calls.

SymPro-LM outperforms few-shot prompting by 21.61, PAL by 3.52 and Logic-LM by 16.83 percent points on FCoReBench, with GPT-4-Turbo as the LLM. Given the structured nature of *fcore* problems, we find that utilizing feedback from small sized solved examples to correct the programs generated for just four rounds yields a further 21.02 percent points gain for SymPro-LM, compared to 12.5 points for PAL.

We further evaluate SymPro-LM on three (non-first order) logical reasoning benchmarks from literature (Tafjord et al., 2021; bench authors, 2023; Saparov and He, 2023a). SymPro-LM consistently outperforms existing baselines by large margins on two datasets, and is competitive on the third, underscoring the value of integrating LLMs with symbolic solvers through programs. We perform additional analyses to understand impact of hyperparameters on SymPro-LM and its errors. We release the dataset and code for further research. We summarize our contributions below:

- We formally define the task of natural language first-order combinatorial reasoning and present FCoReBench, a corresponding benchmark.
- We provide a thorough evaluation of LLM prompting techniques for *fcore* problems, offering new insights into existing techniques.
- We propose a novel approach, SymPro-LM, demonstrating its effectiveness on *fcore* problems as well as other datasets, along with an in-depth analysis of its performance.

## 2 RELATED WORK

**Neuro-Symbolic AI**: Our work falls in the broad category of neuro-symbolic AI (Yu et al., 2023) which builds models leveraging the complementary strengths of neural and symbolic methods. Several prior works build neuro-symbolic models for solving combinatorial reasoning problems (Palm et al., 2018; Wang et al., 2019; Paulus et al., 2021; Nandwani et al., 2022a;b). These develop specialized problem-specific modules (that are typically not size-invariant), which are trained over large training datasets. In contrast, SymPro-LM uses LLMs, and bypasses problem-specific architectures, generalizes to problems of varying sizes, and is trained with very few solved instances.

**Reasoning with Language Models**: The previous paradigm to reasoning was fine-tuning of LLMs (Clark et al., 2021; Tafjord et al., 2021; Yang et al., 2022), but as LLMs scaled, they have been found to reason well, when provided with in-context examples without any fine-tuning (Brown et al., 2020; Wei et al., 2022b). Since then, many prompting approaches have been developed that leverage in-context learning. Prominent ones include Chain of Thought (CoT) prompting (Wei et al., 2022c;

Kojima et al., 2022), Least-to-Most prompting (Zhou et al., 2023), Progressive-Hint prompting (Zheng et al., 2023) and Tree-of-Thoughts (ToT) prompting (Yao et al., 2023).

**Tool Augmented Language Models**: Augmenting LLMs with external tools has emerged as a way to solve complex reasoning problems (Schick et al., 2023; Paranjape et al., 2023). The idea is to offload a part of the task to specialized external tools, thereby reducing error rates. Program-aided Language models (Gao et al., 2023) invoke a Python interpreter over a program generated by an LLM. Logic-LM (Pan et al., 2023) and SAT-LM (Ye et al., 2023) integrate reasoning of symbolic solvers with LLMs, which convert the natural language problem into a symbolic representation. SymPro-LM falls in this category and combines LLMs with *both* program interpreters and symbolic solvers.

**Logical Reasoning Benchmarks**: There are several reasoning benchmarks in literature, such as LogiQA (Liu et al., 2020) for mixed reasoning, GSM8K (Cobbe et al., 2021) for arithmetic reasoning, FOLIO (Han et al., 2022) for first-order logic, PrOntoQA (Saparov and He, 2023b) and ProofWriter (Tafjord et al., 2021) for deductive reasoning, AR-LSAT (Zhong et al., 2021) for analytical reasoning. These dataset are not first-order i.e. each problem is accompanied with a single instance (despite the rules potentially being described in first-order logic). We propose FCoReBench, which substantially extends the complexity of these benchmarks by investigating computationally hard, first-order combinatorial reasoning problems. Among recent works, NLGraph (Wang et al., 2023) studies structured reasoning problems but is limited to graph based problems, and has only 8 problems in its dataset. On the other hand, NPHardEval (Fan et al., 2023) studies problems from the lens of computational complexity, but works with a relatively small set of 10 problems. In contrast we study the more broader area of first-order reasoning, we investigate the associated complexities of structured reasoning, and have a much large problem set (sized 40). Specifically, all the NP-Hard problems in these two datasets are also present in our benchmark.

## 3   PROBLEM SETUP: NATURAL LANGUAGE FIRST-ORDER COMBINATORIAL REASONING

A first-order combinatorial reasoning problem $\mathcal{P}$ has three components: a space of legal input instances ($\mathcal{X}$), a space of legal outputs ($\mathcal{Y}$), and a set of constraints ($\mathcal{C}$) that every input-output pair must satisfy. E.g., for sudoku, $\mathcal{X}$ is the space of partially-filled grids with $n \times n$ cells, $\mathcal{Y}$ is the space of fully-filled grids of the same size, and $\mathcal{C}$ comprises row, column, and box *alldiff* constraints, with input cell persistence. To communicate a structured problem instance (or its output) to an NLP system, it must be serialized in text. We overload $\mathcal{X}$ and $\mathcal{Y}$ to also denote the *formats* for these serialized input and output instances. Two instances for sudoku are shown in Figure 2 (grey box). We are also provided (serialized) training data of input-output instance pairs, $\mathcal{D}_{\mathcal{P}} = \{(x^{(i)}, y^{(i)})\}_{i=1}^{N}$, where $x^{(i)} \in \mathcal{X}, y^{(i)} \in \mathcal{Y}$, such that $(x^{(i)}, y^{(i)})$ honors all constraints in $\mathcal{C}$.

Further, we verbalize all three components – input-output formats and constraints – in natural language instructions. We denote these instructions by $NL(\mathcal{X})$, $NL(\mathcal{Y})$, and $NL(\mathcal{C})$, respectively. Figure 2 illustrates these for su-

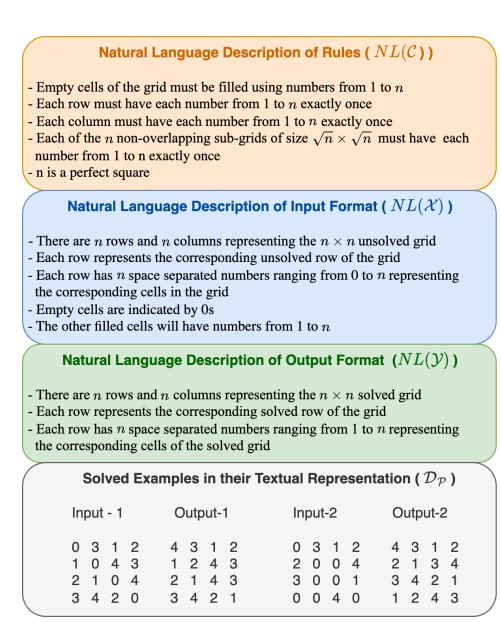

Figure 2: FCoReBench Example: Filling a $n \times n$ Sudoku board along with its rules, input-output format, and a couple of sample input-output pairs.

doku. With this notation, we summarize our setup as follows. For an *fcore* problem $\mathcal{P} = \langle \mathcal{X}, \mathcal{Y}, \mathcal{C} \rangle$, we are provided $NL(\mathcal{X})$, $NL(\mathcal{Y})$, $NL(\mathcal{C})$ and training data $\mathcal{D}_{\mathcal{P}}$, and our goal is to learn a function $\mathcal{F}$, which maps any (serialized) $x \in \mathcal{X}$ to its corresponding (serialized) solution $y \in \mathcal{Y}$ such that $(x, y)$ honors all constraints in $\mathcal{C}$.

## 4 FCoReBench: DATASET CONSTRUCTION

First, we shortlisted computationally challenging first-order problems from various sources. We manually scanned Wikipedia [1] for NP-hard algorithmic problems and logical-puzzles. We also took challenging logical-puzzles from other publishing houses (e.g., Nikoli),[2] and real world problems from the operations research community and the industrial track of the annual SAT competition [2]. From this set, we selected problems (1) that can be described in natural language (we remove problems where some rules are inherently visual), and (2) for whom, the training and test datasets can be created with a reasonable programming effort. This led to 40 *fcore* problems (see Table 7 for a complete list), of which 30 are known to be NP-hard and others have unknown complexity. 10 problems are graph-based (e.g., graph coloring), 18 are grid based (e.g., sudoku), 5 are set-based (e.g., knapsack), 5 are real-world settings (e.g. car sequencing) and 2 are miscellaneous (e.g., cryptarithmetic).

Two authors of the paper having formal background in automated reasoning and logic then created the natural language instructions and the input-output format for each problem. First, for each problem one author created the input-output formats and the instructions for them ($NL(\mathcal{X})$, $NL(\mathcal{Y})$). Second, the same author then created the natural language rules ($NL(\mathcal{C})$) by referring to the respective sources and re-writing the rules. These rules were verified by the other author making sure that they were correct i.e. the meaning of the problem did not change and they were unambiguous. The rules were re-written to ensure that an LLM cannot easily invoke its prior knowledge about the same problem. For the same reason, the name of the problem was hidden.

In the case of errors in the natural language descriptions, feedback was given to the author who wrote the descriptions to correct them. In our case typically there were no corrections required except 3 problems where the descriptions were corrected within a single round of feedback. A third independent annotator was employed who was tasked with reading the natural language descriptions and solving the input instances in the training set. The solutions were then verified to make sure that the rules were written and comprehensible by a human correctly. The annotator was able to solve all instances correctly highlighting that the descriptions were correct. The guidelines utilized to re-write the rules from their respective sources were to use crisp and concise English without utilizing technical jargon and avoiding ambiguities. The rules were intended to be understood by any person with a reasonable comprehension of the language and did not contain any formal specifications or mathematical formulas. Appendices A.2 and A.3 have detailed examples of rules and formats, respectively.

Next, we created train/test data for each problem. These instances are generated programmatically by scripts written by the authors. For each problem, one author also wrote a solver and a verification script, and the other verified that these scripts and suggested corrections if needed. In all but one case the other author found the scripts to be correct. These scripts (after correction) were also verified through manually curated test cases. These scripts were then used to ensure the feasibility of instances.

Since a single problem instance can potentially have multiple correct solutions (Nandwani et al., 2021) – all solutions are provided for each training input. The instances in the test set are typically larger in size than those in training. Because of their size, test instances may have too many solutions, and computing all of them can be expensive. Instead, the verification script can be used, which outputs the correctness of a candidate solution for any test instance. The scripts are a part of the dataset and can be used to generate any number of instances of varying complexity for each problem to easily extend the dataset. Keeping the prohibitive experimentation costs with LLMs in mind, we generate around 15 training instances and around 34 test instances on average per problem. In total FCoReBench has 596 training instances and 1354 test instances.

## 5 SymPro-LM

**Preliminaries**: In the following, we assume that we have access to an LLM $\mathcal{L}$, which can work with various prompting strategies, a program interpreter $\mathcal{I}$, which can execute programs written in its language and a symbolic solver $\mathcal{S}$, which takes as input a pair of the form $(E, V)$, where $E$ is set of

---

[1] https://en.wikipedia.org/wiki/List_of_NP-complete_problems

[2] https://www.nikoli.co.jp/en/puzzles/, https://satcompetition.github.io/

equations (constraints) specified in the language of $\mathcal{S}$, and $V$ is a set of (free) variables in $E$, and produces an assignment $\mathcal{A}$ to the variables in $V$ that satisfies the set of equations in $E$. Given the an *fcore* problem $\mathcal{P} = \langle \mathcal{X}, \mathcal{Y}, \mathcal{C} \rangle$ described by $NL(\mathcal{C})$, $NL(\mathcal{X})$, $NL(\mathcal{Y})$ and $\mathcal{D}_{\mathcal{P}}$, we would like to make effective use of $\mathcal{L}$, $\mathcal{I}$ and $\mathcal{S}$, to learn the mapping $\mathcal{F}$, which takes any input $x \in \mathcal{X}$, and maps it to $y \in \mathcal{Y}$, such that $(x, y)$ honors the constraints in $\mathcal{C}$.

**Background:** We consider the following possible representations for $\mathcal{F}$ which cover existing work.

- **Exclusively LLM**: Many prompting strategies (Wei et al., 2022c; Zhou et al., 2023) make exclusive use of $\mathcal{L}$ to represent $\mathcal{F}$. $\mathcal{L}$ is supplied with a prompt consisting of the description of $\mathcal{P}$ via $NL(\mathcal{C})$, $NL(\mathcal{X})$, $NL(\mathcal{Y})$, the input $x$, along with specific instructions on how to solve the problem and asked to output $y$ directly. This puts the entire burden of discovering $\mathcal{F}$ on the LLM.
- **LLM → Program**: In strategies such as PAL (Gao et al., 2023), the LLM is prompted to output a program, which then is interpreted by $\mathcal{I}$ on the input $x$, to produce the output $y$.
- **LLM + Solver**: Strategies such as Logic-LM (Pan et al., 2023) and Sat-LM (Ye et al., 2023) make use of both the LLM $\mathcal{L}$ and the symbolic solver $\mathcal{S}$. The primary goal of $\mathcal{L}$ is to to act as an interface for translating the problem description for $\mathcal{P}$ and the input $x$, to the language of the solver $\mathcal{S}$. The primary burden of solving the problem is on $\mathcal{S}$, whose output is then parsed as $y$.

## 5.1 OUR APPROACH

Our approach can be seen as a combination of LLM→Program and LLM+Solver strategies described above. While the primary role of the LLM is to do the interfacing between the natural language description of the problem $\mathcal{P}$, the task of solving the actual problem is delegated to the solver $\mathcal{S}$ as in LLM+Solver strategy. But unlike them, where the LLM directly calls the solver, we now prompt it to write a program, $\psi$, which can work with any given input $x \in \mathcal{X}$ of any size. This allows us to get rid of the LLM calls at inference time, resulting in a "lifted" implementation. The program $\psi$ internally represents the specification of the problem. It takes as argument an input $x$, and then converts it according to the inferred specification of the problem to a

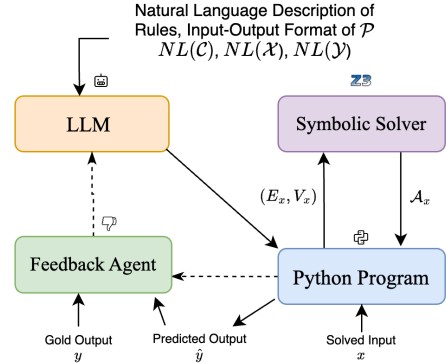

Figure 3: `SymPro-LM`: Solid lines indicate the main flow and dotted lines indicate feedback pathways.

set of equations $(E_x, V_x)$ in the language of the solver $\mathcal{S}$ to get the solution to the original problem. The solver $S$ then outputs an assignment $A_x$ in its own representation, which is then passed back to the program $\psi$, which converts it back to the desired output format specified by $\mathcal{Y}$ and produces output $\hat{y}$. Broadly, our pipeline consists of the 3 components which we describe next in detail.

- **Prompting LLMs**: The LLM is prompted with $NL(\mathcal{C})$, $NL(\mathcal{X})$, $NL(\mathcal{Y})$ (see Figure 2) to generate an input-agnostic program $\psi$. The LLM is instructed to write $\psi$ to read an input from a file, convert it to a symbolic representation according to the inferred specification of the problem, pass the symbolic representation to the solver and then use the solution from the solver to generate the output in the desired format. The LLM is also prompted with information about the solver and its underlying language. Optionally we can also provide the LLM with a subset of $\mathcal{D}_{\mathcal{P}}$ (see Appendix B.3 for exact prompts).
- **Symbolic Solver**: $\psi$ can convert any input instance $x$ to $(E_x, V_x)$ which it passes to the symbolic solver. The solver is agnostic to how the representation $(E_x, V_x)$ was created and tries to find an assignment $A_x$ to $V_x$ which satisfies $E_x$ which is passed back to $\psi$ (see Appendix E.1 for sample programs generated).
- **Generating the Final Output**: $\psi$ then uses $\mathcal{A}_x$ to generate the predicted output $\hat{y}$. This step is need because the symbolic representation was created by $\psi$ and it must recover the desired output representation from $\mathcal{A}_x$, which might not be straightforward for all problem representations.

**Refinement via Solved Examples**: We make use of $\mathcal{D}_{\mathcal{P}}$ to verify and (if needed) make corrections to $\psi$. For each $(x, y) \in \mathcal{D}_{\mathcal{P}}$ (solved input-output pair), we run $\psi$ on $x$ to generate the prediction $\hat{y}$, during which the following can happen: 1) Errors during execution of $\psi$; 2) The solver is unable

to find $\mathcal{A}_x$ under a certain time limit; 3) $\hat{y} \neq y$, i.e. the predicted output is incorrect; 4) $\hat{y} = y$, i.e. the predicted output is correct. If for any training input one of the first three cases occur we provide automated feedback to the LLM through prompts to improve and generate a new program. This process is repeated till all training examples are solved correctly or till a maximum number of feedback rounds is reached. The feedback is simple in nature and includes the nature of the error, the actual error from the interpreter/symbolic solver and the input instance on which the error was generated. For example, in the case where the output doesn't match the gold output we prompt the LLM with the solved example it got wrong and the expected solution. Appendix B contains details of feedback prompts.

It is possible that a single run of `SymPro-LM` (along with feedback) is unable to generate the correct solution for all training examples – so, we restart `SymPro-LM` multiple times for a given problem. Given the probabilistic nature of LLMs a new program is generated at each restart and a new feedback process continues. For the final program, we pick the best program generated during these runs, as judged by the accuracy on the training set. Figure 3 describes our entire approach diagrammatically.

**`SymPro-LM` for Non-First Order Reasoning Datasets**: For datasets that are not first-order in nature, a single program does not exist which can solve all problems, hence we prompt the LLM to generate a new program for each test set instance. Thus we cannot use feedback from solved examples and we only use feedback to correct syntactic mistakes (if any). The prompt contains an instruction to write a program which will use a symbolic solver to solve the problem. Additionally, we provide details about the solver to be used. The prompt also contains in-context examples demonstrating sample programs for other logical reasoning questions. The LLM should parse the logical reasoning question and extract the corresponding facts/rules which it needs to pass to the solver (via the program). Once the solver returns with an answer, it is passed back to the program to generate the final output.

# 6 EXPERIMENTAL SETUP

Our experiments answer these research questions. (1) How does `SymPro-LM` compare with other LLM-based reasoning approaches on *fcore* problems? (2) How useful is using feedback from solved examples and multiple runs for *fcore* problems? (3) How does `SymPro-LM` compare with other methods on other existing (non-first order) logical reasoning benchmarks? (4) What is the nature of errors made by `SymPro-LM` and other baselines?

**Baselines**: On FCoReBench, we compare our method with 4 baselines: 1) *Standard LLM prompting*, which leverages in-context learning to directly answer the questions; 2) *Program-aided Language Models*, which use imperative programs for reasoning and offload the solution step to a program interpreter; 3) *Logic-LM*, which offloads the reasoning to a symbolic solver. 4) *Tree-of-Thoughts* (ToT) Yao et al. (2023), which is a search based prompting technique. These techniques (Yao et al., 2023; Hao et al., 2023) involve considerable manual effort for writing specialized prompts for each problem and are estimated to be 2-3 orders of magnitude more expensive than other baselines. We thus decide to present a separate comparison with ToT on a subset of FCoReBench (see Appendix C.1.1 for more details regarding ToT experiments). We use Z3 (De Moura and Bjørner, 2008) an efficient SMT solver for experiments with Logic-LM and `SymPro-LM`. We use the Python interpreter for experiments with PAL and `SymPro-LM`. We also evaluate *refinement* for PAL and `SymPro-LM` by using 5 runs each with 4 rounds of feedback on solved examples for each problem. We evaluate *refinement* for Logic-LM by providing 4 rounds of feedback to correct syntactic errors in constraints (if any) for each problem instance. We decide not to evaluate SAT-LM given its conceptual similarity to Logic-LM having being proposed concurrently.

**Models**: We experiment with 3 LLMs: GPT-4-Turbo (`gpt-4-0125-preview`) (OpenAI, 2023) which is a SOTA LLM by OpenAI, GPT-3.5-Turbo (`gpt-3.5-turbo-0125`), a relatively smaller LLM by OpenAI and Mixtral 8x7B (`open-mixtral-8x7b`) (Jiang et al., 2024), an open-source mixture-of-experts model developed by Mistral AI. We set the temperature to $0$ for few-shot prompting and Logic-LM for reproducibility and to $0.7$ to sample several runs for PAL and `SymPro-LM`.

**Prompting LLMs**: Each method's prompt includes the natural language description of the problem's rules and the input-output format, along with two solved examples. No additional intermediate supervision (e.g., SMT or Python program) is given in the prompt. For few-shot prompting we directly prompt the LLM to solve each test set instance separately. For PAL we prompt the LLM to write an input-agnostic Python program which reads the input from a file, reasons to solve the

input and then writes the solution to another file, the program generated is run on each testing set instance. For Logic-LM for each test set instance we prompt the LLM to convert it into its symbolic representation which is then fed to a symbolic solver, the prompt additionally contains the description of the language of the solver. We then prompt the LLM with the solution from the solver and ask it to generate the output in the desired format (see Section 5). Prompt templates are detailed in Appendix B and other experimental details can be found in Appendix C.

**Metrics**: For each problem, we use the associated verification script to check the correctness of the candidate solution for each test instance. This script computes the accuracy as the fraction of test instances solved correctly, using binary marking assigning 1 to correct solutions and 0 for incorrect ones. We report the macro-average of test set accuracies across all problems in FCoReBench.

**Additional Datasets**: Apart from FCoReBench, we also evaluate SymPro-LM on 3 additional logical reasoning datasets: (1) *LogicalDeduction* from the BigBench (bench authors, 2023) benchmark, (2) *ProofWriter* (Tafjord et al., 2021) and (3) *PrOntoQA* (Saparov and He, 2023a). In addition to other baselines, we also compare with Chain-of-Thought (CoT) prompting (Wei et al., 2022c), as it performs significantly better than standard prompting for such datasets. Recall that these benchmarks are not first-order in nature i.e. each problem is accompanied with a single instance (despite the rules potentially being first-order) and hence we have to run SymPro-LM (and other methods) separately for each test instance (see Appendix C.2 for more details).

## 7 RESULTS

Table 1 describes the main results for FCoReBench. Unsurprisingly, GPT-4-Turbo is hugely better than other LLMs. Mixtral 8x7B struggles on our benchmark indicating that smaller LLMs (even with mixture of experts) are not as effective at complex reasoning. Mixtral in general does badly, often doing worse than random (especially when used without refinement). PAL and SymPro-LM tend to perform better than other baselines benefiting from the vast pre-training of LLMs on code (Chen et al., 2021). Logic-LM performs rather poorly with smaller LLMs indicating that they struggle to invoke symbolic solvers directly.

Hereafter, we focus primarily on GPT-4-Turbo's performance, since it is far superior to other models. SymPro-LM outperforms few-shot prompting and Logic-LM across all problems in FCoReBench. On average the im-

Table 1: Results for FCoReBench. - / + indicate before / after *refinement*. Performance for random guessing is 20.13%.

| Model | Few-Shot Prompting | PAL | | Logic-LM | | SymPro-LM | |
|---|---|---|---|---|---|---|---|
| | | - | + | - | + | - | + |
| Mixtral 8x7B | 25.06% | 14.98% | 36.09% | 0.21% | 2.04% | 8.08% | 30.09% |
| GPT-3.5-Turbo | 27.02% | 32.66% | 49.19% | 6.04% | 6.58% | 17.08% | 50.35% |
| GPT-4-Turbo | 29.33% | 47.42% | 66.40% | 34.11% | 38.51% | 50.94% | **83.37%** |

provements are by an impressive $54.04\%$ against few-shot prompting and by $44.86\%$ against Logic-LM (with *refinement*). Few-shot prompting solve less than a third of the problems with GPT-4-Turbo, suggesting that even the largest LLMs cannot directly perform complex reasoning. While Logic-LM performs better, it still isn't that good either, indicating that combining LLMs with symbolic solvers is not enough for such reasoning problems.

Table 2: Logic-LM's performance on FCoReBench evaluated with *refinement*.

| Outcome | GPT-3.5-Turbo | GPT-4-Turbo |
|---|---|---|
| Correct Output | 6.58% | 38.51% |
| Incorrect Output | 62.11% | 52.06% |
| Timeout Error | 2.375% | 2.49% |
| Syntactic Error | 29.04% | 6.91% |

Further qualitative analysis suggests that Logic-LM gets confused in handling the structure of *fcore* problems. As problem instance size grows, it tends to make syntactic mistakes with smaller LLMs (Table 2). With larger LLMs, syntactic mistakes reduce, but constraints still remain semantically incorrect and do not get corrected through feedback.

Table 3: Error analysis at a program level for GPT-4-Turbo before and after *refinement* for PAL and SymPro-LM. Results are averaged over all runs for a problem and further over all problems in FCoReBench.

| Outcome | PAL (Before / After) | SymPro-LM (Before / After) |
|---|---|---|
| Incorrect Program | 70% / 57% | 58% / 38% |
| Semantically Incorrect Program | 62% / 49.5% | 29% / 20.5% |
| Python Runtime Error | 7% / 4.5% | 13.5% / 5.5% |
| Timeout | 1% / 3% | 15.5% / 12% |

Often this is because LLMs are error-prone when enumerating combinatorial constraints, i.e., they struggle with executing *implicit* for-loops and conditionals (see Appendix F). In contrast, SymPro-LM and PAL manage first order structures well, since writing code for a loop/conditional is not that hard, and the correct loop-execution is done by a program interpreter. These (size-invariant) programs then get used independently without any LLM call at inference time to solve any input instance – easily generalizing to larger instances – highlighting the benefit of using a program interpreter for such combinatorial problems.

At the same time, PAL is also not as effective on FCoReBench. Table 4 compares the effect of feedback and multiple runs on PAL and SymPro-LM. SymPro-LM outperforms PAL by $16.97\%$ on FCoReBench (with *refinement*). When LLMs are forced to write programs for performing complicated reasoning, they tend to produce brute-force solutions that often are either incorrect or slow (see Table-8 in the appendix). This highlights the value of offloading reasoning to a symbolic solver. Interestingly, feedback from solved examples and re-runs is more effective (Table 3) for SymPro-LM, as also shown by larger gains with increasing number of feedback rounds and runs (Table 4). We hypothesize that this is because declarative programs (generated by SymPro-LM) are easier to correct, than imperative programs (produced by PAL).

Table 4: Comparative analysis between PAL and SymPro-LM on FCoReBench for GPT-4-Turbo.

| | Number of Rounds of Feedback | | | | | | Number of Runs | | | | |
|---|---|---|---|---|---|---|---|---|---|---|---|
| | 0 | 1 | 2 | 3 | 4 | | 1 | 2 | 3 | 4 | 5 |
| PAL | 47.42% | 54.00% | 57.09% | 58.82% | 59.92% | PAL | 59.92% | 62.54% | 63.95% | 65.19% | 66.40% |
| SymPro-LM | 50.94% | 62.54% | 68.52% | 71.12% | 71.96% | SymPro-LM | 71.96% | 77.21% | 80.06% | 82.06% | 83.37% |
| | ↑ 3.52% | ↑ 8.54% | ↑ 11.43% | ↑ 12.3% | ↑ 12.04% | | ↑ 12.04% | ↑ 14.67% | ↑ 16.11% | ↑ 16.87% | ↑ 16.97% |

(a) Effect of feedback rounds for a single run      (b) Effect of multiple runs each with 4 feedback rounds

**Comparison with ToT Prompting**: Table 5 compares SymPro-LM with ToT prompting on 3 problems. SymPro-LM is far superior in terms of cost and accuracy, indicating that even the largest LLMs cannot do complex reasoning on problems with large search depths and branching factors, despite being called multiple times with search-based prompting. Due to its programmatic nature, SymPro-LM generalizes even better to larger instances and is also hugely cost effective, as there is no need to call an LLM for each instance separately. We do not perform further experiments with ToT prompting, due to cost considerations.

Table 5: Accuracy and cost comparison between ToT prompting and SymPro-LM with GPT-4-Turbo for 3 problems in FCoReBench. Costs are per test instance for ToT and one time costs per problem for SymPro-LM.

| Problem | Instance size | ToT prompting | | SymPro-LM | |
|---|---|---|---|---|---|
| | | Accuracy | Cost | Accuracy | Cost |
| Latin Squares | 3x3 | 46.33% | $0.1235 | 100% | $0.02 |
| | 4x4 | 32.5% | $0.5135 | 100% | $0.02 |
| Magic Square | 3x3 | 26.25% | $0.4325 | 100% | $0.02 |
| | 4x4 | 8% | $0.881 | 100% | $0.02 |
| Sujiko | 3x3 | 7.5% | $0.572 | 100% | $0.02 |
| | 4x4 | 0% | $1.676 | 100% | $0.02 |

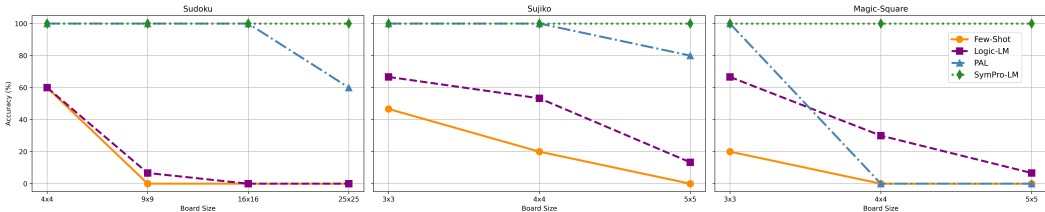

Figure 4: Effect of increasing problem instance size on baselines and SymPro-LM for GPT-4-Turbo.

**Effect of Problem Instance Size**: We now report performance of SymPro-LM and other baselines against varying problem instance sizes (see Figure 4) for 3 problems in FCoReBench (sudoku, sujiko and magic-square). Increasing the problem instance size increases the number of variables, accompanying constraints and reasoning steps required to reach the solution. We observe that being programmatic SymPro-LM and PAL, are relatively robust against increase in size of input instances. In comparison, performance of Logic-LM and few-shot prompting declines sharply. PAL programs are often inefficient and may see performance drop when they fail to find a solution within the time limit.

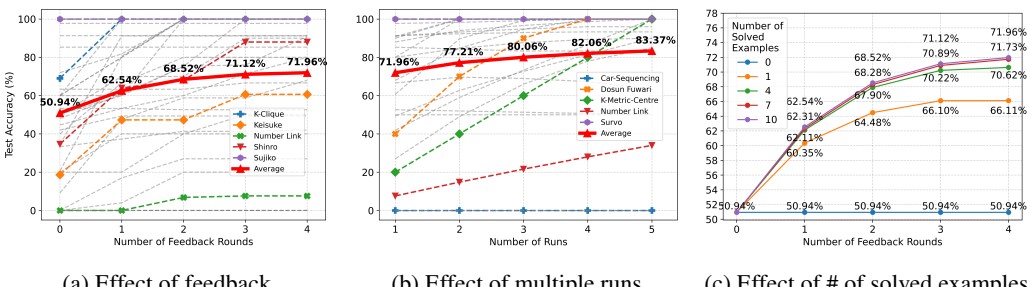

| (a) Effect of feedback | (b) Effect of multiple runs | (c) Effect of # of solved examples |

Figure 5: Effect of feedback and multiple runs with GPT-4-Turbo. (a) and (b) show results with 10 solved examples for feedback where dashed lines show results for individual problems in `FCoReBench`, with coloured lines highlighting specific problems and the red bold line represents the average effect across all problems. (c) shows the effect of number of solved examples used for feedback in a single run.

**Effect of Feedback on Solved Examples**: Figure 5a describes the effect of multiple rounds of feedback for `SymPro-LM`. Feedback helps performance significantly; utilizing 4 feedback rounds improves performance by $21.02\%$. Even the largest LLMs commit errors, making it important to verify and correct their work. But feedback on its own is not enough, a single run might end-up in a wrong reasoning path, which is not corrected by feedback making it important to utilize multiple runs for effective reasoning. Utilizing 5 runs improves the performance by additional $11.41\%$ (Figure 5b) after which the gains tend to saturate. Performance also increases with an increase in the number of solved examples (Figure 5c). Each solved example helps in detecting and correcting different errors. However, performance tends to saturate at 7 solved examples and no new errors are discovered/corrected, even with additional training data.

## 7.1 RESULTS ON OTHER DATASETS

Table 6 reports the performance on non-first order datasets. `SymPro-LM` outperforms all other baselines on ProofWriter and LogicalDeduction, particularly Logic-LM. This showcases the value of integrating LLMs with symbolic solvers through programs, even for standard reasoning tasks. These experiments suggest that LLMs translate natural language questions into programs using solvers much more effectively than into symbolic formulations directly. We attribute this to the vast pre-training of LLMs on code (Brown et al., 2020; Chen et al., 2021). For instance, on the LogicalDeduction benchmark, while Logic-LM does not make syntactic errors during translation it often makes logical errors. These errors significantly decrease when LLMs are prompted to produce programs instead (Figure 6b). Error analysis on ProofWriter and PrOntoQA reveals that for more complex natural language questions, LLMs also start making syntactic errors during translation as the number of rules/facts start increasing. With `SymPro-LM` these errors are vastly reduced because, apart from the benefit from pre-training, LLMs also start utilizing programming constructs like dictionaries and loops to make most out of the structure in these problems (Figure 6a). PAL and CoT perform marginally better on PrOntoQA because the reasoning style for problems in this dataset involves forward-chain reasoning which aligns with PAL's and CoT's style of reasoning. Integrating symbolic solvers is not as useful for this dataset, but still achieves competitive performance.

## 8 DISCUSSION

We analyze `FCoReBench` to identify where LLMs excel and where the largest models still struggle. Based on `SymPro-LM`'s performance, we categorize `FCoReBench` problems into three broad groups.

Table 6: Results for baselines & `SymPro-LM` on other benchmarks. Best results with each LLM are highlighted.

| | GPT-3.5-Turbo-0125 | | | | | GPT-4-Turbo-0125 | | | | |
|---|---|---|---|---|---|---|---|---|---|---|
| Dataset | Direct | CoT | PAL | Logic-LM | SymPro-LM | Direct | CoT | PAL | Logic-LM | SymPro-LM |
| Logical Deduction | 39.66 % | 50.66 % | 66.33 % | 71.00 % | **78.00 %** | 65.33 % | 76.00 % | 81.66 % | 82.67 % | **94.00 %** |
| ProofWriter | 40.50 % | 57.16 % | 50.5 % | 70.16 % | **74.167 %** | 46.5 % | 61.66 % | 76.29 % | 74.83 % | **89.83 %** |
| PrOntoQA | 49.60 % | 83.20 % | **98.40 %** | 72.20 % | 97.40 % | 83.00 % | 98.80 % | **99.80 %** | 91.20 % | 97.80 % |

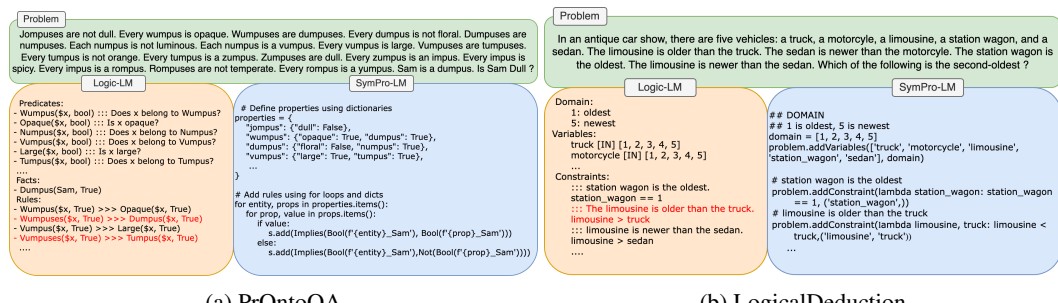

(a) PrOntoQA                       (b) LogicalDeduction

Figure 6: Examples highlighting benefits of integrating LLMs with symbolic solver through programs.

1) Problems that `SymPro-LM` solved with 100% accuracy without any feedback. 8 such problems exist out of the 40, including vertex-cover and latin-square. These problems have a one-to-one correspondence between the natural language description of the rules and the program for generating the constraints and the LLM essentially has to perform a pure translation task which they excel at.

2) Problems that `SymPro-LM` solved with 100% accuracy but after feedback from solved examples. There are 20 such problems. They typically do not have a one-to-one correspondence between rule descriptions and code, thus requiring some reasoning to encode the problem in the solver's language. For eg. one must define auxiliary variables and/or compose several primitives to encode a single natural language rule. GPT-4-Turbo initially misses constraints or encodes the problem incorrectly, but with feedback, it can spot its mistakes and corrects its programs. Examples include k-clique and binairo. In binairo, for example, GPT-4-Turbo incorrectly encodes the constraints for ensuring all columns and rows to be distinct but fixes this mistake after feedback (see Figure 17 in the appendix). LLMs can leverage their vast pre-training to discover non-trivial encodings for several interesting problems and solved examples can help guide LLMs to correct solutions in case of mistakes.

3) Problems with performance below 100% that are not corrected through feedback or utilizing multiple runs. For these 12 problems, LLM finds it difficult to encode some natural language constraint into SMT. Examples include number-link and hamiltonian path, where GPT-4-Turbo is not able to figure out how to encode existence of paths as SMT constraints. In our opinion, these conversions are peculiar, and may be hard even for average CS students. We hope that further analysis of these 12 domains opens up research directions for neuro-symbolic reasoning with LLMs.

## 9   CONCLUSION AND LIMITATIONS

We investigate the reasoning abilities of LLMs on structured first-order combinatorial reasoning problems. We formally define the task, and we present `FCoReBench`, a novel benchmark of 40 such problems and find that existing tool-augmented techniques, such as Logic-LM and PAL fare poorly. In response, we propose `SymPro-LM` – a new technique to aid LLMs with both program interpreters and symbolic solvers. It uses LLMs to convert text into executable code, which is then processed by interpreters to define constraints, allowing symbolic solvers to efficiently tackle the reasoning tasks. Our extensive experiments show that `SymPro-LM`'s integrated approach leads to superior performance on our dataset as well as existing benchmarks. Error analysis reveals that `SymPro-LM` struggles for a certain class of problems where conversion to symbolic representation is not straightforward. In such cases simple feedback strategies do not improve reasoning; exploring methods to alleviate such problems is a promising direction for future work. Another future work direction is to extend this dataset to include images of inputs and outputs, instead of serialized text representations, and assess the reasoning abilities of vision-language models, like GPT4-V.

**Limitations:** While we study a wide variety of *fcore* problems, more such problems always exist and adding these to `FCoReBench` remains a direction of future work. Additionally we assume that input instances and their outputs have a fixed pre-defined (serialized) representation, which may not always be easy to find. Another limitation is that encoding of many problems in the solver's language can potentially be complicated. Our method relies on the pre-training of LLMs to achieve this without any training/fine-tuning, and addressing this is a direction for future work.

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

## A FCoReBench

### A.1 DATASET DETAILS AND STATISTICS

Our dataset namely FCoReBench has $40$ different *fcore* problems that have been collected from various sources. Some of these problems are logical-puzzles from publishing houses like Nikoli, some problems are from operations research literature, some are from the annual SAT competition and other problems are well-known computational problems from Computer Science literature such as hamiltonian path and minimum-dominating set. Table 7 gives the details of all problems in our dataset. To create our training and test sets, we write scripts to synthetically generate problem instances. These can be used to extend the dataset as needed with any number of instances of any size. For experimentation, we generate some solved training instances and a separate set of testing instances. Each problem also has a natural language description of its rules, and a natural language description of the input-format which specify how input problem instances and their solutions are represented in text. The next few sections give illustrative examples and other details.

### A.2 NATURAL LANGUAGE DESCRIPTION OF RULES

This section describes how we create the natural language description of rules for problems in FCoReBench. We extract rules from the sources such as the Wikipedia/Nikoli pages of the corresponding problems. These rules are reworded by a human expert to reduce dataset contamination. Another human expert ensures that there are no ambiguities in the reworded description of the rules. The rules are generalized, when needed (for eg. from a $9 \times 9$ Sudoku to a $n \times n$ Sudoku). The following sections provide few examples.

#### A.2.1 EXAMPLE PROBLEM: SURVO

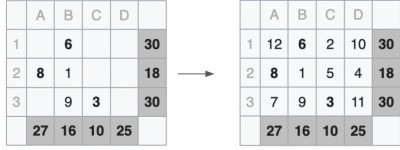

Figure 7: Conversion of an input survo problem instance to its solution.

Survo (Figure 7) is an example problem from FCoReBench. The task is to fill a $m \times n$ rectangular board with numbers from $1 - m * n$ such that each row and column sums to an intended target. (Survo-Wikipedia). The box given below describes the rules of Survo more formally in natural language.

```
We are given a partially filled m × n rectangular board, intended row sums and column sums.
- Empty cells are to be filled with numbers
- Numbers in the solved board can range from 1 to m * n
- Numbers present in filled cells on the input board cannot be removed
- Each number from 1 to m*n must appear exactly once on the solved board
- All the empty cells should be filled such that each row and each column of the solved board
 must sum to the respective row sum and column sum as specified in the input
```

#### A.2.2 EXAMPLE PROBLEM: HAMILTONIAN PATH

Hamiltonian path is a well-known problem in graph theory in which we have to find a path in an un-directed and an un-weighted graph such that each vertex is visited exactly once by the path. We consider the decision variant of this problem which is equally hard in terms of computational complexity. The box below shows the formal rules for this problem expressed in natural language.

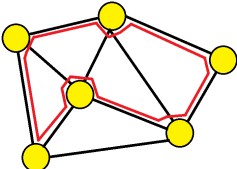

Figure 8: A sample input graph instance and its solution to the hamiltonian-path problem. Vertices are represented by yellow circles and the hamiltonian path is represented by the red line.

```
We are given an un-directed and un-weighted graph.
- We have to determine if the graph contains a path that visits every vertex exactly once.
```

### A.2.3  EXAMPLE PROBLEM: DOSUN FUWARI

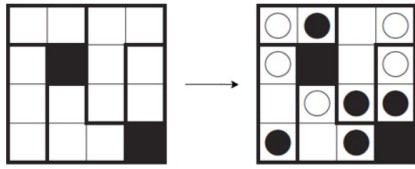

Figure 9: Conversion of an input dosun fuwari problem instance to its solution.

Dosun Fuwari (Nikoli) as shown in Figure 9 is another example problem from FCoReBench. We are given a square board with regions (cells enclosed in bold lines) and we have to fill the board with balloons and iron balls such that one balloon and one iron ball is placed in each region. Balloons are light and float, so they must be placed in one of the cells at the top, in a cell right under a black cell (filled-in cell), or under other balloons. Iron balls are heavy and sink, so they must be placed in one of the cells at the bottom, or in a cell right over a black cell or over other iron balls. The box given below gives the more formal description of the rules of dosun fuwari in natural language.

```
We are given a partially filled n*n square board. We are also given subgrids of the input board.
Cells in the input board can either be empty or filled (that is, nothing can be placed in them,
they are blackened) or can be balloons or iron balls.
- The only thing we can do is place balloons or iron balls in some of or all of the empty cells
- Each subgrid specified in the input should have exactly one balloon and iron ball in the
solved board
- Because balloons are buoyant, they should be positioned either in one of the cells located
at the top of the board or in a cell directly below a filled cell (i.e., one of the blackened
cells in the input) or below other balloons.
- Iron balls, being dense, will sink and should therefore be positioned either directly on one
of the cells located at the bottom of the input board, or on a cell directly above a filled
cell (i.e., one of the blackened cells in the input), or above another iron ball.
```

### A.3  NATURAL LANGUAGE DESCRIPTION OF INPUT AND OUTPUT FORMAT

For many problems we consider input-output instances are typically not represented in text. For each problem we describe a straightforward conversion of the input and output space to text in natural language. The following sections consider examples of a few problems from FCoReBench.

### A.3.1  EXAMPLE PROBLEM: SURVO

Figure 10 represents the conversion of the inputs to survo, originally represented as grid images to text. Here empty cells are denoted by 0's and the filled cells have corresponding values. For a given $m \times n$ board, each row has $m + 1$ space separated integers with the first m integers representing the first row of the input board and the $(m + 1)^{th}$ integer representing the row sum. The last row contains

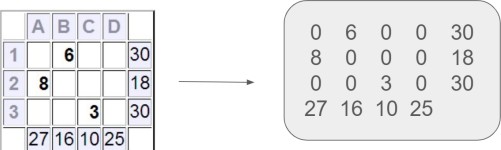

Figure 10: Representation of input instances of survo as text.

$n$ integers represent the column sums. The box below describes this conversion more formally in natural language.

```
Input Format:
- The input will have m + 1 lines
- The first m lines will have n + 1 space-separated integers
- Each of these m lines represents one row of the partially solved input board (n integers),
followed by the required row sum (a single integer)
- The last line of the input will have n space-separated integers each of which represents the
required column sum in the solved board
Sample Input:
0 6 0 0 0 30
8 1 0 0 0 17
0 9 3 0 30
27 16 10 25

Output Format:
- The output should have m lines, each representing one row of the solved board
- Each of these m lines should have n space-separated integers representing the cells of the
solved board
- Each integer should be from 1 to m * n
Sample Output:
12 6 2 10
8 1 5 4
7 9 3 11
```

### A.3.2    EXAMPLE PROBLEM: DOSUN FUWARI

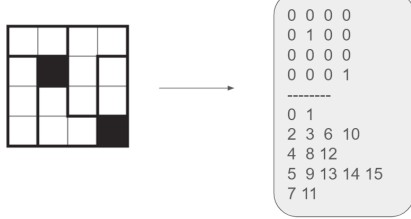

Figure 11: Representation of inputs instances to dosun-fuwari as text.

Figure 11 represents conversion of the inputs to dosun fuwari, originally represented as grid images to text. Here the first few lines represent the input board followed by a string '——' which acts as a separator following which each of the lines has space-separated integers representing the subgrids of the input board. Cells are numbered in row-major order starting from 0, and this numbering is used to represent cells in each of the lines describing the subgrids. In the first few lines representing the input board, 0's represent the empty cells that must be filled. 1's denote the blackened cell, 2s denote the balloons and 3's denote the iron balls. The box below describes these rules more formally in natural language

```
Input-Format:
- The first few lines represent the input board, followed by a line containing ———, which acts
  as a separator, followed by several lines where each line represents one subgrid
- Each of the lines representing the input board will have space-separated integers ranging from
  0 to 3
- 0 denotes empty cells, 1 denotes a filled cell (blackened cell), 2 denotes a cell with a
  balloon, 3 denotes a cell with an iron ball
- After the board, there is a separator line containing ———
- Each of the following lines has space-separated elements representing the subgrids on the
  input board
- Each of these lines has integers representing cells of a subgrid
- Cells are numbered in row-major order starting from 0, and this numbering is used to represent
  cells in each of the lines describing the subgrids

Sample-Input:
0 0 0 0
0 1 0 0
0 0 0 0
0 0 0 1
———
0 1
2 3 6 10
4 8 12
5 9 13 14 15
7 11

Output Format:
- The output should contain as many lines as the size of the input board, each representing one
  row of the solved board
- Each row should have n space separate integers (ranging from 0-3) where n is the size of the
  input board
- Empty cells will be denoted by 0s, filled cells (blackened) by 1s, balloons by 2s and iron
  balls by 3s

Sample-Output:
2 3 0 2
2 1 0 2
0 2 3 3
3 0 3 1
```

### A.3.3 EXAMPLE PROBLEM: HAMILTONIAN PATH

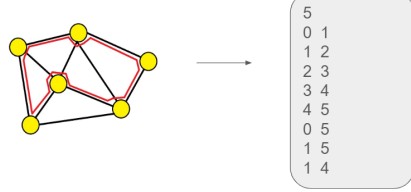

Figure 12: Representation of input instances to hamiltonian-path as text.

Figure 12 represents the conversion of inputs to hamiltonian-path, originally represented as graph image to text. The first line denotes the number of vertices present in the graph followed by which each node of the graph will be numbered from 0 - N-1. Each of the subsequent lines represents an edge of the graph and will contain two space-separated integers (according to the numbering defined previously). The output is a single word (YES/NO) indicating if a hamiltonian path exists in the graph. The box below describes this more formally in natural language.

Input Format:
- The first line will contain a single integer N, the number of nodes in the graph
- The nodes of the graph will be numbered from 0 to N-1
- Each of the subsequent lines will represent an edge of the graph and will contain two space-separated integers (according to the numbering defined above)

Sample-Input:
5
0 1
1 2
2 3
3 4

Output Format:
- The output should contain a single line with a single word
- The word should be YES if a path exists in the input graph according to constraints specified above and NO otherwise

Sample Output:
YES

Table 7: Names of problems in FCoReBench, number of samples in the training set, number of samples in the test set, average size of input instances in training set, average size of input instances in test set and computational complexity. The brackets in the 4th column describe how input instance sizes are measured. ? in the computational complexity column indicates that results are not available for the corresponding problem.

| Problem Name | Training Set Size | Test Set Size | Average Size of Input Instances in Training Set | Average Size of Input Instances in Test Set | Computational Complexity |
|---|---|---|---|---|---|
| 3-Partition (Non Decision) | 15 | 30 | 12 (array size) | 17.7 | NP-Hard |
| 3-Partition (Decision) | 15 | 30 | 12 (array size) | 17.7 | NP-Complete |
| Binario | 15 | 50 | 4.0×4.0 (grid size) | 6.96×6.96 | NP-Hard (De Biasi, 2013) |
| Car-Sequencing | 15 | 30 | 6.96, 3.66, 4.33 (# of cars, # of options, # of classes) | 9.06, 5.66, 6.33 | NP-Hard (Kis, 2004) |
| Clique Cover | 15 | 30 | 6.26, 9.4 (# of nodes, # of edges) | 12.9, 31.4 | NP-Complete |
| Cryptarithmetic | 15 | 30 | 4.32 (Average # of digits in the two operands ) | 4.26 | NP-Hard (Epstein, 1987) |
| Dosun Fuwari | 15 | 30 | 3.066×3.066 (grid size) | 5.23×5.23 | NP-Hard (Iwamoto and Ibusuki, 2018) |
| Futoshiki | 15 | 47 | 5×5 (grid size) | 7.57×7.57 | NP-Hard (Lloyd et al., 2022) |
| Fill-a-pix | 15 | 35 | 2.87 × 2.87 (grid size) | 4.1 × 4.1 | NP-Hard (HIGUCHI and KIMURA, 2019) |
| Flow-Shop | 15 | 30 | 6.06, 3.4 (# of jobs, #num of machines) | 3.83, 9.13 | NP-Complete (Garey et al., 1976a) |
| Factory Workers | 15 | 30 | 5.73, 12.66 (# of factories, # of workers) | 12.35, 30.0 | ? |
| Graph Coloring | 15 | 30 | 5.13, 6.8 (# of nodes, # of edges) | 9, 21.06 | NP-Complete (Gent et al., 2017) |
| Hamiltonian Path | 15 | 30 | 5.93, 8.6 (# of nodes, # of edges) | 13.0, 19.77 | NP-Complete |
| Hamiltonian Cycle | 15 | 30 | 5.93, 8.6 (# of nodes, # of edges) | 11.07, 18.67 | NP-Complete |
| Hidato | 15 | 45 | 2.87 × 2.87 (grid size) | 4.1 × 4.1 | NP-Hard (Itai et al., 1982) |
| Independent Set | 12 | 30 | 5.8, 7.2 (# of nodes, # of edges) | 14.2, 29.8 | NP-Complete |
| Inshi-No-Heya | 15 | 49 | 5.0×5.0 (grid size) | 6.5×6.5 | ? |
| Job-Shop | 15 | 30 | 3.66, 3.66 (# of jobs, # of machines) | 9, 9 | NP-Complete (Garey et al., 1976b) |
| K-Clique | 15 | 31 | 4.87, 7.6 (# of nodes, # of edges) | 8.84, 26.97 | NP-Complete |
| Keisuke | 15 | 30 | 4.33×4.33 (grid size) | 5.83×5.83 | ? |
| Ken Ken | 15 | 20 | 3.26×3.26 (grid size) | 5.2×5.2 | NP-Hard (Haraguchi and Ono, 2015) |
| Knapsack | 15 | 30 | 4.8 (array size) | 24.56 | NP-Hard |
| K Metric Centre | 15 | 30 | 4.5 (# of nodes) | 7 | NP-Hard |
| Latin Square | 15 | 50 | 6×6.0 (grid size) | 14.3×14.3 | NP-Hard (Colbourn, 1984) |
| Longest Path Problem | 15 | 30 | 6.2, 5.87 (# of nodes, # of edges) | 12.6, 16.3 | NP-Complete |
| Magic Square | 15 | 30 | 3.0×3.0 (grid size) | 4.33×4.33 | ? |
| Minimum Dominating Set | 15 | 30 | 6.0, 17.73 (# of nodes, # of edges) | 14.53, 45.0 | NP-Complete |
| N-Queens | 15 | 30 | 3.8×3.8 (grid size) | 6.33×6.33 | NP-Hard (Gent et al., 2017) |
| Number Link | 15 | 50 | 4×4 (grid size) | 7.1×7.1 | NP-Hard |
| Partition Problem | 15 | 35 | 7.06 (array size) | 15 | NP-Complete |
| PRP | 15 | 30 | 4.93, 12.6 (# of units, # of days) | 6.7, 23.9 | ? |
| Shinro | 15 | 30 | 5.13×5.13 (grid size) | 9.2×9.2 | ? |
| Subset Sum | 15 | 30 | 3.67 (array size) | 11.87 | NP-Complete |
| Summle | 15 | 20 | 2.33 (# of equations) | 3.75 | ? |
| Sudoku | 15 | 50 | 4.0×4.0 (grid size) | 13.3×13.3 | NP-Hard (YATO and SETA, 2003) |
| Sujiko | 15 | 45 | 3.0×3.0 (grid size) | 4.0×4.0 | ? |
| Survo | 15 | 47 | 13.5 (area of grid) | 20.25 | ? |
| Symmetric Sudoku | 15 | 30 | 4×4 (grid size) | 6.5×6.5 | ? |
| Sliding Tiles | 15 | 30 | 2.66 × 2.66, 6.13 (grid size, search depth) | 3.63 × 3.63, 8.83 | NP-Complete (Demaine and Rudoy, 2018) |
| Vertex Cover | 14 | 30 | 6.4, 13.4 (# of nodes, # of edges) | 12.6, 40.4 | NP-Complete |

## B  PROMPT TEMPLATES

In this section we provide prompt templates used for our experiments on FCoReBench, including the templates for the baselines we experimented with, SymPro-LM as well as prompt templates for providing feedback.

### B.1  FEW-SHOT PROMPT TEMPLATE

```
Task:
<Description of the Rules of the problems>

Input-Format:
<Description of Textual Representation of Inputs>
<Input Few Shot Example-1>
<Input Few Shot Example-2>
........................
........................
<Input Few Shot Example-n>

Output-Format
<Description of Textual Representation of Outputs>

<Output of Few Shot Example-1>
<Output of Few Shot Example-2>
...........................
...........................
<Output of Few Shot Example-n>

Input problem instance to be solved:
<Problem Instance from the Test Set>
```

### B.2  PAL PROMPT TEMPLATE

The following box describes the base prompt template used for PAL experiments with FCoReBench.

```
Write a Python program to solve the following problem:

Task:
<Description of the Rules of the problem>

Input-Format:
<Description of Textual Representation of Inputs>
Sample-Input:
<Sample Input from Feedback Set>

Output-Format:
<Description of Textual Representation of Outputs>
Sample-Output:
<Output of Sample Input from Feedback Set>

Don't write anything apart from the Python program; use Python comments if needed.

The Python program is expected to read the input from input.txt and write the output to a file
named output.txt.

The Python program must only use standard Python libraries.
```

### B.3  SymPro-LM TEMPLATE

#### B.3.1  BASE PROMPT

```
  Write a Python program to solve the following problem:

  Task:
  <Description of the Rules of the problem>

  Input-Format:
  <Description of Textual Representation of Inputs>
  Sample-Input:
  <Sample Input from Feedback Set>

  Output-Format:
  <Description of Textual Representation of Outputs>
  Sample-Output:
  <Output of Sample Input from Feedback Set>

  The Python program must read the input from input.txt and convert that particular input to the
  corresponding constraints, which it should pass to the Z3 solver, and then it should use the Z3
  solver's output to write the solution to a file named output.txt

  Don't write anything apart from the Python program; use Python comments if needed.
```

## B.4    FEEDBACK PROMPT TEMPLATES

These prompt templates are used to provide feedback in the case of SymPro-LM or PAL.

### B.4.1    PROGRAMMING ERRORS

```
Your code is incorrect and produces the following runtime error:<RUN TIME ERROR> for the following
input: <INPUT> rewrite your code and fix the mistake
```

### B.4.2    VERIFICATION ERROR

```
Your code is incorrect, when run on the input: <INPUT> the output produced is <OUTPUT-GENERATED>
which is incorrect whereas one of the correct output is <GOLD-OUTPUT>.
Rewrite your code and fix the mistake.
```

### B.4.3    TIMEOUT ERROR

```
Your code was inefficient and took more than <TIME-LIMIT> seconds to execute for the following input:
<INPUT>.
Rewrite the code and optimize it.
```

## B.5 LOGIC-LM PROMPT TEMPLATE

The following box describes the prompt for Logic-LM experiments with FCoReBench, the prompt is used to convert the input to its symbolic representation.

```
Task:
<Description of the Rules of the problem>

Input-Format:
<Description of Textual Representation of Inputs>
Sample-Input:
<Sample Input from Feedback Set>

Output-Format:
<Description of Textual Representation of Outputs>
Sample-Output:
<Output of Sample Input from Feedback Set>

Input problem to be solved:
<Problem Instance from the Test Set>

The task is to declare variables and the corresponding constraints on them in SMT2 for the
input mentioned above. The variables and constraints should be such that once the variables are
solved for, one can use the solution to the variables (which satisfies the constraints) to get
to the output in the desired format for the above mentioned input.

Only Write the SMT2 code and nothing else.  Write the complete set of SMT2 variables
and constraints. Enclose SMT2 code in "`smt2 "`
```

## B.6 TOT

In this section we give an example of the ToT prompts used for experiments on FCoReBench. We use latin square as the running example.

### B.6.1 PROPOSE PROMPT

This prompt is called for each search node to get the possible next states.

```
Task:
We are given a nxn partially solved board and have to solve it according to the following rules:
- We need to replace the 0s with numbers from 1-n.
- Non-zero numbers on the board cannot be replaced.
- Each number from 1-n must appear exactly once in each column and row in the solved board
Given a board, decide which cell to fill in next and the number to fill it with, each possible
next step is separated by a new line.
You can output up-to 3 next steps.
If the input board is fully filled or no valid next step exists output only 'END'.

Sample-Input-1:
1 0 3
2 0 0
0 1 2
Possible next steps for Sample Input-1:
1 2 3
2 0 0
0 1 2

1 0 3
2 0 0
3 1 2

1 0 3
2 3 0
0 1 2

Sample-Input-2:
1 2 3
2 3 1
3 1 2
Possible next steps for Sample Input-2:
END

Input:
<node from the search tree>
Possible next steps for Input:
```

## B.6.2   VALUE PROMPT

This prompt is called for each search node to evaluate how likely it is to get to the solution from that
node. We use this to prune the search tree.

```
Task:
We are given a nxn partially solved board and have to solve it according to the following rules:
- We need to replace the 0s with numbers from 1-n.
- Non-zero numbers already on the board cannot be replaced.
- Each number from 1-n must appear exactly once in each column and row in the solved board.
Given a partially filled board, evaluate how likely it is to reach a valid solution
(sure/likely/impossible)

Output-Format:
The output should have two lines as follows:
<Reasoning>
<Sure/Likely/Impossible>
Sample-Input-1:
0 0 0
0 0 0
0 0 0
Board is empty, hence it is always possible to get to a solution.
Sure

Sample-Input-2:
1 0 3
2 0 0
0 1 2
No constraint is violated till now and it is likely to get to a solution.
Likely

Sample-Input-3:
1 1 3
2 0 0
0 1 2
Constraint violated in first row.
Impossible

Input:
<node from the search tree>
```

## C Experimental Details

### C.1 FCoReBench

All methods are evaluated zero-shot, meaning no in-context demonstrations for the task are provided to the LLM. We choose the zero-shot setting for FCoReBench because of the structured nature of problems, making it unfair to provide demonstrations of highly related problems instances to the LLM. The LLM is only given a description of the rules of the problem and the task it has to perform. For PAL and SymPro-LM we present results with 10 solved examples for feedback.

### C.1.1 ToT prompting

We evaluate ToT prompting (Yao et al., 2023) on 3 problems in FCoReBench. Our implementation closely resembles the official implementation which we adapt for grid based logical puzzles. We use a BFS based approach with propose and value prompts. An example prompt for latin square can be found in Appendix B.6. Problems in our benchmark have huge branching factors, to reduce experimentation cost, we greedily prune the search frontier to 5 nodes at each depth based on scores assigned by the LLM during the value stage. Additionally during the propose stage we prompt the LLM to output at most 3 possible next steps. The temperature is set to 0.0 for reproducibility. Unlike the original implementation problems in our benchmark can have varying search depths, hence we explicitly ask the LLM to output 'END' once a terminal node is reached. At any depth if a terminal node is amongst the best nodes we terminate the search and return the terminal nodes at that depth, otherwise we search till a maximum search depth governed by the problem instance.

## C.2 Other Datasets

We evaluate `SymPro-LM` on 3 other datasets apart from `FCoReBench`. Our evaluation closely follows Logic-LM's evaluation (Pan et al., 2023). For baselines we use the same prompts as Logic-LM. Logic-LM did not evaluate PAL, for which we write prompts on our own similar to the CoT prompts used by Logic-LM. For `SymPro-LM` we write prompts on our own. We use the same in-context examples as used for Logic-LM. We instruct the LLM to write a Python program to parse the input problem, setup variables/constraints and pass these to a symbolic solver, call the solver and using the solver's output print the final answer. For LogicalDeduction we use the `python-constraints` [3] package which is a CSP solver. For other datasets we use the Z3-solver [4]. Since all problems are single correct MCQ questions we use accuracy as our metric. Like Logic-LM if there is an error during execution of the program generated by the LLM we fall back on using chain-of-thought to predict the answer. The following sections provide descriptions for the datasets used.

### C.2.1 PRONTOQA

PrOntoQA (Saparov and He, 2023a) is a recent synthetic dataset created to analyze the deductive reasoning capacity of LLMs. We use the hardest fictional characters version of the dataset, based on the results in (Saparov and He, 2023a). Each version is divided into different subsets depending on the number of reasoning hops required. We use the hardest 5-hop subset for evaluation. Each question in PrOntoQA aims to validate a new fact's veracity, such as "True or false: Alex is not shy." The following box provides an example:

```
Context: Each jompus is fruity. Every jompus is a wumpus. Every wumpus is not transparent.
Wumpuses are tumpuses. Tumpuses are mean. Tumpuses are vumpuses. Every vumpus is cold. Each
vumpus is a yumpus. Yumpuses are orange. Yumpuses are numpuses. Numpuses are dull. Each numpus
is a dumpus. Every dumpus is not shy. Impuses are shy. Dumpuses are rompuses. Each rompus is
liquid. Rompuses are zumpuses. Alex is a tumpus

Question: True or false: Alex is not shy.
Options:
A) True
B) False
```

### C.2.2 PROOFWRITER

ProofWriter (Tafjord et al., 2021) is another commonly used dataset for deductive logical reasoning. Compared with PrOntoQA, the problems are expressed in a more naturalistic language form. We use the open-world assumption (OWA) subset in which each example is a (problem, goal) pair and the label is one of PROVED, DISPROVED, UNKNOWN. The dataset is divided into five parts each part requiring $0, \leq 1, \leq 2, \leq 3$, and $\leq 5$ hops of reasoning, respectively. We evaluate the hardest depth-5 subset. To reduce overall experimentation costs, we randomly sample 600 examples in the test set and ensure a balanced label distribution. The following box provides an example:

```
Context: Anne is quiet. Erin is furry. Erin is green. Fiona is furry. Fiona is quiet.
Fiona is red. Fiona is rough. Fiona is white. Harry is furry. Harry is quiet. Harry
is white. Young people are furry. If Anne is quiet then Anne is red. Young, green
people are rough. If someone is green then they are white. If someone is furry and quiet
then they are white. If someone is young and white then they are rough. All red people are young.

Question: Based on the above information, is the following statement true, false, or
unknown? Anne is white.
Options:
A) True
B) False
C) Uncertain
```

---

[3] https://github.com/python-constraint/python-constraint
[4] https://pypi.org/project/z3-solver/

### C.2.3 LOGICALDEDUCTION

LogicalDeduction bench authors, 2023 is a challenging logical reasoning task from the BigBench collaborative benchmark. The problems are mostly about deducing the order of a sequence of objects from a minimal set of conditions. We use the full test set consisting of 300 examples. The following box provides an example:

```
Context: The following paragraphs each describe a set of three objects arranged in a fixed
order. The statements are logically consistent within each paragraph. In an antique car show,
there are three vehicles: a station wagon, a convertible, and a minivan. The station wagon is
the oldest. The minivan is newer than the convertible.

Question: Which of the following is true?
Options:
A) The station wagon is the second-newest.
B) The convertible is the second-newest.
C) The minivan is the second-newest.
```

### C.3 HARDWARE DETAILS

All experiments were conducted on an Intel(R) Xeon(R) Gold 6226R CPU @ 2.90GHz, 32 cores, 64-bit, with 512 KiB L1 cache, 16 MiB L2 cache, and 22 MiB L3 cache. We accessed GPT-4-Turbo and GPT-3.5-Turbo by invoking both models via the OpenAI API. Mixtral 8x7B was also accessed by using the Mistral AI API although the model weights are available publicly. We preferred the API, over running the model locally given the ease of setup because all our other experiments were with APIs.

## D ADDITIONAL RESULTS

### D.1 INFERENCE TIME

The following tables describes the average inference time for test set instances of a few illustrative problems in FCoReBench. SymPro-LM performs much better compared to PAL because PAL programs

| Problem | PAL | SymPro-LM |
|---|---|---|
| Sudoku | 2.01 | 0.215 |
| Latin Square | 5.46 | 0.2 |
| Cryptarithmetic | 0.83 | 0.73 |
| Independent Set | 1.438 | 0.106 |
| Minimum Dominating Set | 0.98 | 0.112 |
| Sujiko | 0.742 | 0.102 |
| Vertex Cover | 1.58 | 0.105 |

Table 8: Average inference time in seconds of SymPro-LM and PAL for test set instances for selected problems in FCoReBench

often tend to be brute force and inefficient whereas the solver can exploit the nature of the input-instance while performing the reasoning with SymPro-LM.

# E EXAMPLES

## E.1 SymPro-LM

### E.1.1 FCoReBench

This section includes example programs generated by SymPro-LM for some illustrative problems in FCoReBench. Each program reads the input from a file, generates the corresponding constraints, calls the solver internally and then uses the solution from the solver to write the output in the desired format to a file.

```python
from z3 import Solver, Int, And, Distinct, Or
import math

# Function to read the board from the file
def read_board(filename):
    with open(filename, 'r') as file:
        board = [[int(num) for num in line.split()] for line in file]
    return board

# Function to write the solved board to the file
def write_board(filename, board):
    with open(filename, 'w') as file:
        for row in board:
            file.write(' '.join(map(str, row))+'\n')

# Function to solve the Sudoku-like puzzle
def solve_puzzle(board):
    n = len(board)
    sqrt_n = int(math.sqrt(n))
    s = Solver()

    # Create a matrix of integer variables
    X = [[Int(f'X_{i+1}_{j+1}') for j in range(n)] for i in range(n)]

    # Constraints for each cell to be between 1 and n
    cells_c = [And(1 <= X[i][j], X[i][j] <= n) for i in range(n) for j in range(n)]

    # Constraints for rows and columns to be distinct
    rows_c = [Distinct(X[i]) for i in range(n)]
    cols_c = [Distinct([X[i][j] for i in range(n)]) for j in range(n)]

    # Constraints for subgrids to be distinct
    subgrids_c = [Distinct([X[i][j]
                    for i in range(k*sqrt_n, (k+1)*sqrt_n)
                        for j in range(l*sqrt_n, (l+1)*sqrt_n)])
                            for k in range(sqrt_n) for l in range(sqrt_n)]

    # Constraints for the given cells to match the input
    given_c = [X[i][j] == board[i][j] for i in range(n) for j in range(n) if board[i][j] != 0]

    # Add all constraints to the solver
    s.add(cells_c + rows_c + cols_c + subgrids_c + given_c)

    if s.check() == 'sat':
        m = s.model()
        solved_board = [[m.evaluate(X[i][j]).as_long() for j in range(n)] for i in range(n)]
        return solved_board
    else:
        return None

# Main process
if _name_ == '_main_':
    input_filename = 'input.txt'
    output_filename = 'output.txt'

    board = read_board(input_filename)
    solved_board = solve_puzzle(board)

    if solved_board:
        write_board(output_filename, solved_board)
    else:
        print("Failed to solve the puzzle")
```

Figure 13: SymPro-LM example: correct program for sudoku generated by GPT-4-Turbo.

```python
from z3 import *

def parse_input(file_path):
    with open(file_path, 'r') as file:
        lines = [line.strip() for line in file.readlines()]
    board = [list(line.split()) for line in lines[:-2]]
    across_nums = lines[-2].split()
    down_nums = lines[-1].split()
    return board, across_nums, down_nums

def solve_crossword(board, across_nums, down_nums):
    size = len(board)
    solver = Solver()

    # Create a 2D array of Z3 integer variables
    X = [[Int(f"x_{i}_{j}") for j in range(size)] for i in range(size)]

    # Apply constraints based on the initial board
    for i in range(size):
        for j in range(size):
            if board[i][j] == "-":
                solver.add(X[i][j] == -1)
            elif board[i][j] != "0":
                solver.add(X[i][j] == int(board[i][j]))
            else:
                solver.add(And(X[i][j] >= 0, X[i][j] <= 9))

    # Function to generate constraints for placing numbers
    def place_numbers(numbers, vertical=False):
        for num in numbers:
            num_len = len(num)
            placements = []
            for i in range(size):
                for j in range(size):
                    if vertical:
                        if i + num_len <= size:
                            seq = [X[i+k][j] == int(num[k]) for k in range(num_len)]
                            placements.append(And(*seq))
                    else:
                        if j + num_len <= size:
                            seq = [X[i][j+k] == int(num[k]) for k in range(num_len)]
                            placements.append(And(*seq))
            solver.add(Or(*placements))

    place_numbers(across_nums, vertical=False)
    place_numbers(down_nums, vertical=True)

    if solver.check() == sat:
        model = solver.model()
        solution = [[model.evaluate(X[i][j]) for j in range(size)] for i in range(size)]
        return solution
    else:
        return None

def write_output(file_path, solution):
    with open(file_path, 'w') as file:
        for row in solution:
            line = ' '.join('-' if val.as_long() == -1 else str(val.as_long()) for val in row)
            file.write(f"{line}\n")

def main():
    input_file = 'input.txt'
    output_file = 'output.txt'
    board, across_nums, down_nums = parse_input(input_file)
    solution = solve_crossword(board, across_nums, down_nums)
    if solution:
        write_output(output_file, solution)
    else:
        print("No solution found.")

if __name__ == "__main__":
    main()
```

Figure 14: SymPro-LM example: correct program for keisuke generated by GPT-4-Turbo.

```python
from z3 import *

def read_input(filename):
    with open(filename, 'r') as file:
        n, k = map(int, file.readline().split())
        edges = [tuple(map(int, line.split())) for line in file]
    return n, k, edges

def write_output(filename, answer):
    with open(filename, 'w') as file:
        file.write(answer)

def solve_graph_path(n, k, edges):
    s = Solver()
    path = [Int(f'path_{i}') for i in range(k+1)] # Representing vertices along the path

    # Every vertex in the path must be within graph vertices
    for vertex in path:
        s.add(vertex >= 0, vertex < n)

    # Edges constraint
    for i in range(k):
        s.add(Or([And(path[i] == edge[0], path[i+1] == edge[1])
                  for edge in edges] +
                 [And(path[i] == edge[1], path[i+1] == edge[0])
                  for edge in edges]))

    # Making sure all vertices in the path are distinct
    for i in range(k+1):
        for j in range(i+1, k+1):
            s.add(path[i] != path[j])

    if s.check() == sat:
        return "YES"
    return "NO"

def main():
    n, k, edges = read_input("input.txt")
    answer = solve_graph_path(n, k, edges)
    write_output("output.txt", answer)

if __name__ == "__main__":
    main()
```

Figure 15: SymPro-LM example: correct program for hamiltonian path generated by GPT-4-Turbo.

```
from z3 import *

# Function to read the graph and the integer K from the file input.txt
def read_input(filename):
    with open(filename, 'r') as file:
        n, k = map(int, file.readline().split())
        edges = [tuple(map(int, line.split())) for line in file]
    return n, k, edges

# Function to check if it's possible to select at most K vertices
def is_possible(n, k, edges):
    # Initialize Z3 Solver
    solver = Solver()

    # Create a list of boolean variables representing whether a vertex is selected
    vertices = [Bool(f'vertex_{i}') for i in range(n)]

    # Add constraint that at most K vertices can be selected
    solver.add(Sum([If(v, 1, 0) for v in vertices]) <= k)

    # For each edge, add constraint that at least one of the vertices should be selected
    for u, v in edges:
        solver.add(Or(vertices[u], vertices[v]))

    # Check if the problem is solvable with the given constraints
    if solver.check() == sat:
        return "YES"
    else:
        return "NO"

# Function to write the output to a file
def write_output(filename, result):
    with open(filename, 'w') as file:
        file.write(result)

# Main function
def main():
    n, k, edges = read_input('input.txt')
    result = is_possible(n, k, edges)
    write_output('output.txt', result)

if __name__ == "__main__":
    main()
```

Figure 16: `SymPro-LM` example: correct program for vertex-cover generated by GPT-4-Turbo.

```
...

### Constraint for ensuring distinct rows/columns is incorrect
### This constraint ensures that each row/column has distinct elements which over-constraints the
    problem.
# Constraints for rows and columns to be distinct
for i in range(n):
    solver.add(Distinct([grid[i][j] for j in range(n)]))  # Rows distinct
    solver.add(Distinct([grid[j][i] for j in range(n)]))  # Columns distinct

...
```

```
...
# Enforcing distinct rows and columns by ensuring sum of differences is non-zero
for i in range(n):
    for j in range(i + 1, n):
        # Ensure row i and row j are different
        solver.add(Sum([If(grid[i][k] != grid[j][k], 1, 0) for k in range(n)]) > 0)
        # Ensure column i and column j are different
        solver.add(Sum([If(grid[k][i] != grid[k][j], 1, 0) for k in range(n)]) > 0)

...
```

Figure 17: `SymPro-LM` example: snippet of incorrect program for binairo generated by GPT-4-Turbo and same snippet after correction by feedback.

```python
from z3 import *

# Function to write the solution to the output file
def write_solution_to_file(solution):
    with open("output.txt", "w") as f:
        if solution is None:
            f.write("None")
        else:
            f.write(" ".join(map(str, solution)))

# Function to solve the problem with Z3
def solve_with_z3(array, target_sum):
    solver = Solver()

    # Create a boolean variable for each element in the array to indicate if it is included in the
        sum
    include_vars = [Bool(f"include_{i}") for i in range(len(array))]

    # Add constraint that the sum of the selected elements should be equal to the target sum
    solver.add(Sum([If(include_vars[i], array[i], 0) for i in range(len(array))]) == target_sum)

    # Try to find a solution
    if solver.check() == sat:
        model = solver.model()
        # Extract the solution: which elements to include
        solution = [array[i] for i in range(len(array)) if is_true(model[include_vars[i]])]
        return solution
    else:
        return None

# Main function to read the input, solve the problem, and write the output
def main():
    with open("input.txt", "r") as f:
        lines = f.read().strip().split("\n")
        array = list(map(int, lines[0].split()))
        target_sum = int(lines[1])

    solution = solve_with_z3(array, target_sum)
    write_solution_to_file(solution)

# Execute the main function
if __name__ == "__main__":
    main()
```

Figure 18: SymPro-LM example: correct program for subset-sum generated by GPT-4-Turbo.

## E.1.2 OTHER DATASETS

```python
from z3 import *

# Define properties using dictionaries
properties = {
    "jompus": {"fruity": True, "wumpus": True},
    "wumpus": {"transparent": False, "tumpus": True},
    "tumpus": {"mean": True, "vumpus": True},
    "vumpus": {"cold": True, "yumpus": True},
    "yumpus": {"orange": True, "numpus": True},
    "numpus": {"dull": True, "dumpus": True},
    "dumpus": {"shy": False, "rompus": True},
    "impus": {"shy": True},
    "rompus": {"liquid": True, "zumpus": True},
}

# Create Z3 solver
s = Solver()

# Declare the individual
Alex = Bool('Alex')

# Add facts
s.add(Bool('tumpus_Alex'))

# Add rules using for loops and dicts
for entity, props in properties.items():
    for prop, value in props.items():
        if value:
            s.add(Implies(Bool(f'{entity}_Alex'), Bool(f'{prop}_Alex')))
        else:
            s.add(Implies(Bool(f'{entity}_Alex'), Not(Bool(f'{prop}_Alex'))))

# Define Alex as a tumpus
s.add(Bool('tumpus_Alex'))

# Query if Alex is not shy
s.add(Not(Bool('shy_Alex')))

# Check the solution
answer = 'A' if s.check() == sat else 'B'
```

Figure 19: SymPro-LM PrOntaQA Example Program.

```python
from z3 import *

s = Solver()

### Define predicates as Z3 Booleans
Quiet = Function('Quiet', IntSort(), BoolSort())
Furry = Function('Furry', IntSort(), BoolSort())
Green = Function('Green', IntSort(), BoolSort())
Red = Function('Red', IntSort(), BoolSort())
Rough = Function('Rough', IntSort(), BoolSort())
White = Function('White', IntSort(), BoolSort())
Young = Function('Young', IntSort(), BoolSort())

### Define persons as integers
Anne, Erin, Fiona, Harry = Ints('Anne Erin Fiona Harry')

### Add facts to the solver
s.add(Quiet(Anne))
s.add(Furry(Erin), Green(Erin))
s.add(Furry(Fiona), Quiet(Fiona), Red(Fiona), Rough(Fiona), White(Fiona))
s.add(Furry(Harry), Quiet(Harry), White(Harry))

# Add rules to the solver
def add_rules(solver):
    x = Int('x')
    solver.add(ForAll(x, Implies(Young(x), Furry(x)))) # Young people are furry
    solver.add(Implies(Quiet(Anne), Red(Anne))) # If Anne is quiet then Anne is red
    solver.add(ForAll(x, Implies(Or(Young(x), Green(x)), Rough(x))))  # Young, green people are rough
    solver.add(ForAll(x, Implies(Green(x), White(x))))  # If someone is green then they are white
    solver.add(ForAll(x, Implies(And(Furry(x), Quiet(x)), White(x)))) # If someone is furry and quiet
            then they are white
    solver.add(ForAll(x, Implies(And(Young(x), White(x)), Rough(x))))  # If someone is young and
            white then they are rough
    solver.add(ForAll(x, Implies(Red(x), Young(x))))  # All red people are young
add_rules(s)

query = White(Anne) # Query: Is Anne white?
# Check if the negation of the query leads to a contradiction
s.push()  # Create a new context to test the negation
s.add(Not(query))

answer = ''
if s.check() == unsat:
    answer = "A"
else:
    s.pop()  # Restore the previous state
    s.add(query)  # Directly add the query now
    if s.check() == unsat:
        answer = "B"
    else:
        answer = "C"
```

Figure 20: SymPro-LM ProofWriter Example Program.

```python
from constraint import *

problem = Problem()
## DOMAIN
## 1 is oldest
## 3 is newest
domain = [1, 2, 3]
problem.addVariables(['station_wagon', 'convertible', 'minivan'], domain)

### CONSTRAINTS
problem.addConstraint(lambda station_wagon: station_wagon == 1, ('station_wagon',)) # station wagon
        is the oldest
problem.addConstraint(lambda minivan, convertible: minivan > convertible, ('minivan', 'convertible'))
        # minivan is newer than the convertible
problem.addConstraint(lambda station_wagon, convertible, minivan: len({station_wagon, convertible,
        minivan}) == 3, ('station_wagon', 'convertible', 'minivan'))  # Ensure all vehicles have
        different ages

## SOLUTION
answer = ""
solution = problem.getSolutions()[0]
if solution["station_wagon"] == 2: # A) The station wagon is the second-newest.
    answer = "A"
elif solution["convertible"] == 2: # B) The convertible is the second-newest.
    answer = "B"
elif solution["minivan"] == 2: # C) The minivan is the second-newest.
    answer = "C"
```

Figure 21: SymPro-LM LogicalDeduction Example Program.

## E.2  PAL

This section includes example programs generated by PAL for some illustrative problems in FCoReBench. Each program reads the input from a file, performs the reasoning and writes the output to another text file.

```python
import math

def read_board(filename):
    with open(filename, 'r') as file:
        board = [[int(num) for num in line.split()] for line in file]
    return board

def write_board(filename, board):
    with open(filename, 'w') as file:
        for row in board:
            file.write(' '.join(map(str, row)) + '\n')

def is_valid(board, row, col, num):
    n = len(board)
    box_size = int(math.sqrt(n))
    row_start = (row // box_size) * box_size
    col_start = (col // box_size) * box_size

    if any(num == board[row][i] for i in range(n)):
        return False
    if any(num == board[i][col] for i in range(n)):
        return False
    if any(num == board[i][j] for i in range(row_start, row_start + box_size) for j in range(
            col_start, col_start + box_size)):
        return False
    return True

def solve(board):
    n = len(board)
    for row in range(n):
        for col in range(n):
            if board[row][col] == 0:
                for num in range(1, n+1):
                    if is_valid(board, row, col, num):
                        board[row][col] = num
                        if solve(board):
                            return True
                        board[row][col] = 0
                return False
    return True

def main():
    board = read_board('input.txt')
    if solve(board):
        write_board('output.txt', board)
    else:
        print("No solution exists.")

if __name__ == "__main__":
    main()
```

Figure 22: PAL example: correct program for sudoku generated by GPT-4-Turbo.

```python
import itertools

def read_input(filename):
    """Read the input file and return the board and constraints."""
    with open(filename, "r") as file:
        lines = file.readlines()

    n = len(lines[0].strip().split())
    board = [list(map(int, line.strip().split())) for line in lines[:n]]
    constraints = [tuple(map(int, line.strip().split())) for line in lines[n:]]

    return board, constraints

def write_output(filename, board):
    """Write the solution to the output file."""
    with open(filename, "w") as file:
        for row in board:
            file.write(' '.join(map(str, row)) + '\n')

def check_constraints(board, constraints):
    """Check if the board satisfies all the constraints."""
    n = len(board)
    for x, y in constraints:
        x_row, x_col = divmod(x, n)
        y_row, y_col = divmod(y, n)
        if board[x_row][x_col] >= board[y_row][y_col]:
            return False
    return True

def solve_board(board, constraints):
    """Solve the board with the given constraints."""
    n = len(board)
    rows = [set(range(1, n + 1)) - set(row) for row in board]
    cols = [set(range(1, n + 1)) - set(board[row][col] for row in range(n)) for col in range(n)]

    def backtrack(row, col):
        if row == n:
            return check_constraints(board, constraints)
        if col == n:
            return backtrack(row + 1, 0)
        if board[row][col] == 0:
            for num in rows[row] & cols[col]:
                board[row][col] = num
                rows[row].remove(num)
                cols[col].remove(num)

                if backtrack(row, col + 1):
                    return True

                board[row][col] = 0
                rows[row].add(num)
                cols[col].add(num)
        else:
            return backtrack(row, col + 1)
        return False

    backtrack(0, 0)
    return board

def main():
    input_filename = 'input.txt'
    output_filename = 'output.txt'

    board, constraints = read_input(input_filename)
    solved_board = solve_board(board, constraints)
    write_output(output_filename, solved_board)

if __name__ == "__main__":
    main()
```

Figure 23: PAL example: correct program for futoshiki generated by GPT-4-Turbo.

```python
def dfs(graph, node, visited, N, path_length):
    if path_length == N:
        # If all nodes have been visited, we found a Hamiltonian path
        return True
    for neighbor in graph[node]:
        if not visited[neighbor]:
            visited[neighbor] = True
            if dfs(graph, neighbor, visited, N, path_length + 1):
                return True
            visited[neighbor] = False
    return False

def has_hamiltonian_path(N, edges):
    # Initialize the graph as an adjacency list
    graph = {i: [] for i in range(N)}
    for edge in edges:
        graph[edge[0]].append(edge[1])
        graph[edge[1]].append(edge[0])

    # Try to find Hamiltonian path starting from each node
    for start_node in range(N):
        visited = [False] * N
        visited[start_node] = True
        if dfs(graph, start_node, visited, N, 1):
            return True
    return False

def main():
    # Read the input file
    with open('input.txt', 'r') as file:
        lines = file.readlines()

    # Parse the number of nodes
    N = int(lines[0].strip())

    # Parse the edges
    edges = [tuple(map(int, line.strip().split())) for line in lines[1:]]

    # Check if a Hamiltonian path exists
    path_exists = has_hamiltonian_path(N, edges)

    # Write the output to a file
    with open('output.txt', 'w') as file:
        file.write('YES' if path_exists else 'NO')

# Execute the program
if __name__ == '__main__':
    main()
```

Figure 24: PAL example: correct program for hamiltonian path generated by GPT-4-Turbo.

```
from itertools import combinations

# Function to read the graph from a file
def read_graph(file_path):
    with open(file_path, 'r') as file:
        lines = file.readlines()
    N, K = map(int, lines[0].split())
    edges = [tuple(map(int, line.split())) for line in lines[1:]]
    return N, K, edges

# Function to check if a set of vertices is a valid cover
def is_valid_cover(vertices, edges):
    for edge in edges:
        if edge[0] not in vertices and edge[1] not in vertices:
            return False
    return True

# Loading the graph
N, K, edges = read_graph('input.txt')

# Attempting to find a valid cover with at most K vertices
found = False
for num_vertices in range(1, K+1):
    for vertices in combinations(range(N), num_vertices):
        if is_valid_cover(vertices, edges):
            found = True
            break
    if found:
        break

# Writing the result to the file
with open('output.txt', 'w') as file:
    file.write('YES' if found else 'NO')
```

Figure 25: PAL example: correct program for vertex cover generated by GPT-4-Turbo.

```
def find_combinations_that_sum_to_target(array, target, start=0, path=[]):
    """
    This function finds combinations in 'array' that sum to 'target'.
    It returns a list of all possible combinations that sum up to the target.
    """
    if target == 0:
        return [path]
    if target < 0:
        return []
    combinations = []
    for i in range(start, len(array)):
        if i > start and array[i] == array[i-1]:
            continue
        combinations += find_combinations_that_sum_to_target(array, target-array[i], i+1, path+[array
            [i]])
    return combinations

# Read input from input.txt
with open('input.txt', 'r') as file:
    input_lines = file.readlines()
    array = list(map(int, input_lines[0].strip().split()))
    target = int(input_lines[1].strip())

# Find combinations
combinations = find_combinations_that_sum_to_target(array, target)

# Write output to output.txt
with open('output.txt', 'w') as file:
    if combinations:
        # Just write the first combination found to the output file
        file.write(' '.join(map(str, combinations[0])))
    else:
        file.write('None')
```

Figure 26: PAL example: correct program for subset sum generated by GPT-4-Turbo.

# F  LOGIC-LM

This section describes example runs of Logic-LM for certain problems in FCoReBench.

```
"""Input Instance"""
## 0 4 3 2
## 2 0 1 0
## 0 0 0 0
## 0 0 2 3

; Declare the variables
(declare-const a11 Int)
(declare-const a12 Int)
(declare-const a13 Int)
(declare-const a14 Int)
(declare-const a21 Int)
(declare-const a22 Int)
(declare-const a23 Int)
(declare-const a24 Int)
(declare-const a31 Int)
(declare-const a32 Int)
(declare-const a33 Int)
(declare-const a34 Int)
(declare-const a41 Int)
(declare-const a42 Int)
(declare-const a43 Int)
(declare-const a44 Int)

; Initialize known values
(assert (= a12 4))
(assert (= a13 3))
(assert (= a14 2))
(assert (= a21 2))
(assert (= a23 1))
(assert (= a43 2))
(assert (= a44 3))

; Each cell must be between 1 and 4
(assert (and (>= a11 1) (<= a11 4)))
(assert (and (>= a22 1) (<= a22 4)))
(assert (and (>= a24 1) (<= a24 4)))
(assert (and (>= a31 1) (<= a31 4)))
(assert (and (>= a32 1) (<= a32 4)))
(assert (and (>= a33 1) (<= a33 4)))
(assert (and (>= a34 1) (<= a34 4)))
(assert (and (>= a41 1) (<= a41 4)))
(assert (and (>= a42 1) (<= a42 4)))

; Each number must appear exactly once in each row
(assert (distinct a11 a12 a13 a14))
(assert (distinct a21 a22 a23 a24))
(assert (distinct a31 a32 a33 a34))
(assert (distinct a41 a42 a43 a44))

; Each number must appear exactly once in each column
(assert (distinct a11 a21 a31 a41))
(assert (distinct a12 a22 a32 a42))
(assert (distinct a13 a23 a33 a43))
(assert (distinct a14 a24 a34 a44))

; Each number must appear exactly once in each 2x2 subgrid
(assert (distinct a11 a12 a21 a22))
(assert (distinct a13 a14 a23 a24))
(assert (distinct a31 a32 a41 a42))
(assert (distinct a33 a34 a43 a44))

; Check and get the model
(check-sat)
(get-model)
```

Figure 27: Logic-LM example: correct constraints for a sudoku instance generated by GPT-4-Turbo.

```
"""Input Instance"""
# 1 2 3 4 5 6 7 8 9 10
# 55

(declare-const a1 Bool)
(declare-const a2 Bool)
(declare-const a3 Bool)
(declare-const a4 Bool)
(declare-const a5 Bool)
(declare-const a6 Bool)
(declare-const a7 Bool)
(declare-const a8 Bool)
(declare-const a9 Bool)
(declare-const a10 Bool)

(assert (= (+ (ite a1 1 0) (ite a2 2 0) (ite a3 3 0) (ite a4 4 0) (ite a5 5 0) (ite a6 6 0) (ite a7 7
      0) (ite a8 8 0) (ite a9 9 0) (ite a10 10 0)) 55))

(check-sat)
(get-model)
```

Figure 28: Logic-LM example: correct constraints for a subset sum instance generated by GPT-4-Turbo.

```
"""Input Instance"""
# 8 3
# 0 1
# 0 2
# 2 4
# 3 4
# 3 7
# 6 7
# 0 6
# 5 1
# 5 0
# 5 2
# 5 4
# 5 3
# 5 7
# 5 6

; Declare colour variables for each vertex
(declare-const c0 Int)
(declare-const c1 Int)
(declare-const c2 Int)
(declare-const c3 Int)
(declare-const c4 Int)
(declare-const c5 Int)
(declare-const c6 Int)
(declare-const c7 Int)

; Constraint: Each colour variable must be in the range [0, K-1], where K=3 for this problem
(assert (and (>= c0 0) (< c0 3)))
(assert (and (>= c1 0) (< c1 3)))
(assert (and (>= c2 0) (< c2 3)))
(assert (and (>= c3 0) (< c3 3)))
(assert (and (>= c4 0) (< c4 3)))
(assert (and (>= c5 0) (< c5 3)))
(assert (and (>= c6 0) (< c6 3)))
(assert (and (>= c7 0) (< c7 3)))

; Constraint: Adjacent vertices must have different colours
(assert (not (= c0 c1)))
(assert (not (= c0 c2)))
(assert (not (= c2 c4)))
(assert (not (= c3 c4)))
(assert (not (= c3 c7)))
(assert (not (= c6 c7)))
(assert (not (= c0 c6)))
(assert (not (= c5 c1)))
(assert (not (= c5 c0)))
(assert (not (= c5 c2)))
(assert (not (= c5 c4)))
(assert (not (= c5 c3)))
(assert (not (= c5 c7)))
(assert (not (= c5 c6)))

; Check if there is a solution
(check-sat)
```

Figure 29: Logic-LM example: correct constraints for graph coloring instance generated by GPT-4-Turbo.

```
"""Input Instance"""
# 0 1 3 0
# 4 3 2 1
# 0 4 1 0
# 1 0 0 0
(declare-const a11 Int)
(declare-const a12 Int)
(declare-const a13 Int)
(declare-const a14 Int)

(declare-const a21 Int)
(declare-const a22 Int)
(declare-const a23 Int)
(declare-const a24 Int)

(declare-const a31 Int)
(declare-const a32 Int)
(declare-const a33 Int)
(declare-const a34 Int)

(declare-const a41 Int)
(declare-const a42 Int)
(declare-const a43 Int)
(declare-const a44 Int)

(assert (and (>= a11 1) (<= a11 4)))
(assert (and (>= a12 1) (<= a12 4)))
(assert (and (>= a13 1) (<= a13 4)))
(assert (and (>= a14 1) (<= a14 4)))

(assert (and (>= a21 1) (<= a21 4)))
(assert (and (>= a22 1) (<= a22 4)))
(assert (and (>= a23 1) (<= a23 4)))
(assert (and (>= a24 1) (<= a24 4)))

(assert (and (>= a31 1) (<= a31 4)))
(assert (and (>= a32 1) (<= a32 4)))
(assert (and (>= a33 1) (<= a33 4)))
(assert (and (>= a34 1) (<= a34 4)))

(assert (and (>= a41 1) (<= a41 4)))
(assert (and (>= a42 1) (<= a42 4)))
### SYNTAX ERROR
(assert (>= a43 1) (<= a43 4)))
### SYNTAX ERROR
(assert (and (>= a44 1) (<= a44 4)))

(assert (distinct a11 a12 a13 a14))
(assert (distinct a21 a22 a23 a24))
(assert (distinct a31 a32 a33 a34))
(assert (distinct a41 a42 a43 a44))

(assert (distinct a11 a21 a31 a41))
(assert (distinct a12 a22 a32 a42))
(assert (distinct a13 a23 a33 a43))
(assert (distinct a14 a24 a34 a44))

(assert (distinct a11 a12 a21 a22))
(assert (distinct a13 a14 a23 a24))
(assert (distinct a31 a32 a41 a42))
(assert (distinct a33 a34 a43 a44))

(assert (distinct a11 a12 a13 a14))
(assert (distinct a21 a22 a23 a24))
(assert (distinct a31 a32 a33 a34))
(assert (distinct a41 a42 a43 a44))

(check-sat)
(get-model)
```

Figure 30: Logic-LM example: syntax error (highlighted by a comment) in constraints for sudoku instance generated by GPT-3.5-Turbo.

```
"""Input Instance"""
# 1 0 0 2
# 4 0 1 0
# 2 0 0 1
# 3 0 2 0

; Declare the variables
(declare-const a11 Int)
(declare-const a12 Int)
(declare-const a13 Int)
(declare-const a14 Int)
(declare-const a21 Int)
(declare-const a22 Int)
(declare-const a23 Int)
(declare-const a24 Int)
(declare-const a31 Int)
(declare-const a32 Int)
(declare-const a33 Int)
(declare-const a34 Int)
(declare-const a41 Int)
(declare-const a42 Int)
(declare-const a43 Int)
(declare-const a44 Int)

; Initialize known values
(assert (= a11 1))
###  ERROR
(assert (= a13 1))
###  ERROR
(assert (= a21 4))
###  ERROR
(assert (= a23 2))
###  ERROR
(assert (= a31 2))
(assert (= a34 1))
(assert (= a41 3))
(assert (= a43 2))

; Each cell must be between 1 and 4
(assert (and (>= a12 1) (<= a12 4)))
(assert (and (>= a14 1) (<= a14 4)))
(assert (and (>= a22 1) (<= a22 4)))
(assert (and (>= a24 1) (<= a24 4)))
(assert (and (>= a32 1) (<= a32 4)))
(assert (and (>= a33 1) (<= a33 4)))
(assert (and (>= a42 1) (<= a42 4)))
(assert (and (>= a44 1) (<= a44 4)))

; Each number must appear exactly once in each row
(assert (distinct a11 a12 a13 a14))
(assert (distinct a21 a22 a23 a24))
(assert (distinct a31 a32 a33 a34))
(assert (distinct a41 a42 a43 a44))

; Each number must appear exactly once in each column
(assert (distinct a11 a21 a31 a41))
(assert (distinct a12 a22 a32 a42))
(assert (distinct a13 a23 a33 a43))
(assert (distinct a14 a24 a34 a44))

; Each number must appear exactly once in each 2x2 subgrid
(assert (distinct a11 a12 a21 a22))
(assert (distinct a13 a14 a23 a24))
(assert (distinct a31 a32 a41 a42))
(assert (distinct a33 a34 a43 a44))

; Check and get the model
(check-sat)
(get-model)
```

Figure 31: Logic-LM example: errors (highlighted comments) in constraints for sudoku instance generated by GPT-4-Turbo.