# OpenReview forum: "FCoReBench: Can Large Language Models Solve Challenging First-Order Combinatorial Reasoning Problems?"
_ICLR.cc/2025/Conference — Submitted to ICLR 2025_

### Official Review · Reviewer_3HGQ · 2024-10-27

**Soundness:** 2
**Presentation:** 2
**Contribution:** 2
**Rating:** 3
**Confidence:** 4

**Summary:**

This paper introduces a problem set designed to assess LLMs' ability to solve first-order combinatorial reasoning problems. It argues that current symbolic-solver-aided LLMs perform poorly on this problem set and proposes a novel approach that combines a symbolic solver with a program interpreter to improve reasoning capabilities, demonstrating superior performance on the problems.

**Strengths:**

The paper aims to address an important problem. The proposed approach is conceptually sound, and the experimental results indicate promising improvements in the reasoning capabilities of LLMs when using the technique.

**Weaknesses:**

This paper has several critical issues that require the authors' attention:

1. Misalignment Between Title and Content: While the title suggests a focus on the proposed problem set, the main body primarily discusses the technique, SymPro-LM. After reviewing the entire paper, it appears more as a technique paper rather than a benchmark paper. I suggest revising the title and reorganizing the structure to more accurately reflect its focus on methodology.

2. Lack of Clarity on Incremental Contributions of the Problem Set: Although the problem set seems useful, the paper does not clearly articulate its unique contribution. Existing symbolic-solver-aided LLM approaches have already addressed similar reasoning problems, and some may have been tested on benchmarks containing first-order combinatorial reasoning problems. It is essential to compare the proposed problem set with these existing benchmarks, highlighting overlaps and differences. However, this paper provides limited detail on this aspect.

3. Scope Restriction and Generalizability of the Technique: While the paper narrows its focus to first-order combinatorial reasoning problems, conceptually, the proposed technique has broader applicability across various reasoning tasks. Given the absence of any domain-specific adaptations, I recommend either expanding the paper’s scope and conducting a more comprehensive evaluation across diverse reasoning problems, or explaining the reason of the scope restriction.

4. Use of an Outdated LLM in Evaluation: The LLM used in the evaluations appears a bit outdated. I suggest incorporating recent models, such as GPT-4o and o1, to provide a more relevant assessment.

5. Unclear Criteria for Problem Selection in the Problem Set: The criteria for including specific problems in the problem set are not well-defined. For example, while the paper includes problems from the industry track of SAT competitions, it does not explain the exclusion of others (e.g., the main track). Furthermore, recent SAT competitions no longer feature an industry track, making the rationale for this selection unclear.

**Questions:**

What is the criteria of problem selection?
What is the overlap and difference between your problem set and existing problem set?
Why does the paper restrict its scope within first-order combinatorial reasoning problems?

---

### Official Review · Reviewer_8qGB · 2024-10-29

**Soundness:** 3
**Presentation:** 3
**Contribution:** 3
**Rating:** 5
**Confidence:** 3

**Summary:**

The paper introduces FCoReBench, a benchmark designed to evaluate the capabilities of LLMs in solving first-order combinatorial reasoning problems. The benchmarks include NP-hard problem instances like graph coloring and knapsack, with varying instance sizes. Current LLMs struggle with these tasks, particularly as the problem size increases. To address this limitation, this paper proposes SymPro-LM, a hybrid approach that combines LLMs with symbolic solvers, enhancing performance by leveraging the strengths of both methods.

The proposed approach achieved a 21.61% improvement over few-shot prompting, a 3.52% improvement over Program-aided Language models (PAL), and a 16.83% enhancement over Logic-LM. Additionally, incorporating feedback from solved examples boosts SymPro-LM's performance by 21.02% after four rounds, compared to 12.5% for PAL. SymPro-LM also excels on three non-first-order logical reasoning benchmarks, outperforming existing baselines on two datasets and remaining competitive on the third, highlighting the effectiveness of integrating LLMs with symbolic solvers.

**Strengths:**

Using LLM to solve logic puzzles and combinatorial problems is a very important and interesting direction. This paper contributes a well-established dataset for this field which can be valuable to the research community. The paper also proposes a framework that combines extant solvers such as Z3 with LLMs. The experiment results seem convincing and promising.

**Weaknesses:**

1. The name "first order" is a bit confusing. Does it mean it is related to first-order logic? If so, it would be great to elaborate on this connection. Otherwise, a more detailed definition should be provided. It is not clear from the paper what the difference is between first-order problems and second-order ones.

2. Whether the level of contribution of this paper meets the standard of ICLR is questionable. It is not clear whether this paper proposed novel methodologies. The main contribution according to the paper seems to be the establishment of a dataset.

Minor: The fonts in Figure 4 should be larger.

**Questions:**

What is the difference between first-order and non-first-order problems?

---

### Official Review · Reviewer_fByT · 2024-11-02

**Soundness:** 2
**Presentation:** 1
**Contribution:** 2
**Rating:** 3
**Confidence:** 3

**Summary:**

Introduces FCoReBench which consists of generators and evaluators for 40 combinatorial optimization problems such as sudoku, graph coloring etc. Evaluates existing prompting approaches and LLM augmentation approaches on the dataset. Proposes a new framework SymPro-LLM which when given a problem, output a program that converts the problem to symbolic representation, which is then passed to a symbolic solver to get the solution.

**Strengths:**

The proposed SymPro-LLM can work with different instances from the same first order combinatorial optimization problem without the need to re-evaluate using LLMs.

The proposed dataset is difficult for existing LLMs. The instances are based on combinatorial reasoning problems, which are mostly NP-Hard problems.

The proposed dataset is lifted such that unlimited new instances can be generated.

**Weaknesses:**

While I find the proposed approach of using LLM to output program to formulate models interesting, I am not convinced the experimentations conducted provide enough insight to LLM reasoning abilities. From the examples shown in figure 2, NL(C), NL(X), NL(Y) seems to be pseudo code for formulating the problem. The task of the LLM therefore becomes translating the pseudo code to python, which does not require the same level of reasoning as solving the problems.

The paper does not evaluate enough existing models for the new proposed benchmark dataset. For example, the state-of-the-art GPT-4o and GPT-o1 are not evaluated. The paper also include limited analysis of why the existing approaches fail on the proposed dataset.

The writing and presentation require more clarity and focus. For example, section 7 presents results across different LLM models, different frameworks/styles of prompting, different datasets/problem classes, and different experimental setups. It is unclear to me what the key takeaways from these results are.

**Questions:**

Is there a possibility of data contamination, where the LLMs have seen these combinatorial optimization problems in their training data, and therefore know how to formulate them easily?

Do you have further insights on why Logic-LM performance is so much worse in Table 1? It is also formulating the model and offloading the reasoning to a solver similar to the proposed framework.

Table 1: Considering the NP-Hard nature of the problems, how come random guess achieves over 20% accuracy? What are your expectations for the eventual performance of LLMs on this dataset?

---

### Official Review · Reviewer_yJYh · 2024-11-04

**Soundness:** 2
**Presentation:** 1
**Contribution:** 2
**Rating:** 3
**Confidence:** 2

**Summary:**

This paper focuses on the problem-solving ability of LLM on first-order combinatorial problems in natural language form, arguing that no existing benchmark could reveal this challange properly. To stress the significance of this issue, this paper proposes a new benchmark, FCoReBench, which covers 40 challanging problems in varying sizes and correspounding solutions. In responding to the poor performance of current LLMs on FCoReBench, this paper further proposes a new framework, SymPro-LM, to push forward the potential capacity of language models by combining symbolic solvers, program interpreters and the LM backbone. The experimental results show a significant improvement in various aspects, indicating the valuable attempt of assembling different augmented modules.

**Strengths:**

- The problems covered in FCoReBench are relatively comprehensive, highlighting a valuable research direction. It would be interesting to see more generalized problems to be addressed once VLM are taken into consideration.
- A corresponding responce framework has been developed for the issue proposed, and the experimental results are promising.
- The experimental section in section 7 features thorough verification and comprehensive chart presentations.
- The discussion in section 8 is insightful. It would be benificial to list the problems in each situation in the appendix, and even better, to illustrated them with diagrams in the main text. This would help to elucidate the dataset's relevance to the central issue.

**Weaknesses:**

- The construction part of the dataset issue in Section 4 requires manual labor, which is quite labor-intensive. Could it be automated using LLM?
- The current agent can only solve first-order logic. Higher-order logic requires individual generation, which is resource-intensive and difficult to scale.
- There is a lack of innovation in the proposed framework SymPro-LM, which merely combines existing symbolic solvers and program generation. It would be better to consider a more specific design.

Writing aspects:
- There are issues with the section layout and organization; the section titles are inconsistent and not uniformly formatted (e.g. section 5 and 5.1, section 7 and 7.1). The table layout on page 7 is also peculiar.
- The overall language used in writing is subpar, being rather colloquial and informal. E.g.:
  - In Section 3, as a problem definition, there should not be such an emphasis on the subject "We." The problem should be described objectively and rigorously from a third-party perspective.
  - In Section 4, the term "the author" should be used less frequently to avoid potential privacy issues. Instead, use "agent" or "process" to emphasize actions rather than the actors, which would be more formal. If necessary, flowcharts can also be used to represent the selection, polishing, and construction processes, which would greatly assist readers in understanding the overall procedures.
- This paper primarily focuses on the benchmark, as emphasized in the title; thus, the experimental section should mainly focus on verifying the performance of the benchmark in various aspects. The current writing approach is centered around SymPro-LM. If this focus is to be maintained, the emphasis of the entire article should be placed on SymPro-LM.

**Questions:**

- I do not understand the sentence in page 3, line 122: "These dataset are not first-order i.e. each problem is accompanied with a single instance (despite the rules potentially being described in first-order logic)."
- In page 3 line 151, does the training data $\mathcal{D}_\mathcal{P}$ condition on the previous $\mathcal{C}$ given by different problems? In my understanding, different $(x, y)$ pair may have different $\mathcal{C}$.
- In page 4 line 179-181: "The rules were re-written to ensure that an LLM cannot easily invoke its prior knowledge about the same problem. For the same reason, the name of the problem was hidden." Why can't let LLM be aware of the given problem catagory?
- In page 4 line 191: "did not contain any formal specifications or mathematical formulas" I don't know why this rule is set.
- In the last paragraph of section 4, why does the instances in the test set are typically larger in size than those in training? Also, why does the number of instances in test dataset smaller than those in the train dataset?
- In section 5.1, what is the "training" object of SymPro-LM? More specifically, this framework does not need any nn training. What does it optimize during iterations?
- In page 6 line 318, why do you use different temperature on different solvers? Could you explain the detailed reason?
- In page 7 line 349, what is the "random" approach here?
- Some typos:
  - In page 3 line 149: "We are also provided" -> "We also provides"
  - In page 4 line 213: "which takes as input" -> "which takes input as"
  - In page 5 line 266: "This step is need" -> "This step is needed"
  - In page 10 line 504: "but after" -> "only after"

---

### Meta-Review · Area_Chair_WMQX · 2024-12-22

**Metareview:**

This paper introduces a new benchmark, FCoReBench, consisting of 40 combinatorial optimization problems whose constraints, inputs, outputs, and examples are all stated in natural languages. Additionally, this paper also proposes a new framework, SymPro-LM, which outperforms existing prompting methods like few-shot prompting and program-aided prompting.  Introducing new datasets and prompting frameworks to improve the reasoning capability of LLMs are valuable contributions. However, several important concerns are not addressed properly. For instance, to what extent the 40 combinatorial problems are new compared to existing reasoning tasks? Although the problem is stated in natural language, the form is still rigid -- clear separations regarding constraints, I/O instructions, and examples have to be specified, making it not far from a piece of pseudo-code. Furthermore, the key motivation is not very clear; on one hand, it suggests the contribution of benchmark, the novelty of which is a bit questionable; on the other hand, the authors want to show the new framework, SymPro-LM, significantly outperforms existing techniques on a newly crafted benchmark. A more systematic comparison of existing benchmarks where baseline approaches were evaluated would be expected.

**Additional Comments On Reviewer Discussion:**

As the authors did not respond, there was unfortunately no discussion.

---

### Decision · Program_Chairs · 2025-01-22

Reject